# Dual-level Mixup for Graph Few-shot Learning with Fewer Tasks

## Abstract

Graph neural networks have been demonstrated as a powerful paradigm for effectively learning graph-structured data on the *web* and mining content from it. Current leading graph models require a large number of labeled samples for training, which unavoidably leads to overfitting in few-shot scenarios. Recent research has sought to alleviate this issue by simultaneously leveraging graph learning and meta-learning paradigms. However, these graph meta-learning models assume the availability of numerous meta-training tasks to learn transferable meta-knowledge. Such assumption may not be feasible in the real world due to the difficulty of constructing tasks and the substantial costs involved. Therefore, we propose a **SiM**ple yet effect**I**ve approach for graph few-shot **L**earning with f**E**wer tasks, named **SMILE**. We introduce a dual-level mixup strategy, encompassing both within-task and across-task mixup, to simultaneously enrich the available nodes and tasks in meta-learning. Moreover, we explicitly leverage the prior information provided by the node degrees in the graph to encode expressive node representations. Theoretically, we demonstrate that SMILE can enhance the model generalization ability. Empirically, SMILE consistently outperforms other competitive models by a large margin across all evaluated datasets with in-domain and cross-domain settings. Our anonymous code can be found here.

## CCS Concepts

• **Information systems → Data mining**; • **Computing methodologies → Neural networks**.

## Keywords

Graph neural network, Few-shot learning, Node classification

**ACM Reference Format:**
Anonymous Author(s). 2018. Dual-level Mixup for Graph Few-shot Learning with Fewer Tasks. In *Proceedings of Make sure to enter the correct conference title from your rights confirmation emai (Conference acronym 'XX).* ACM, New York, NY, USA, 22 pages. https://doi.org/XXXXXXX.XXXXXXX

## 1 Introduction

As a fundamental data structure, graphs can effectively model complex relationships between objects and they are ubiquitous in the real world. Graph neural networks (GNNs) have been widely employed as an effective tool for graph task analysis [12, 18, 22, 47, 56]. Prevailing GNN models are designed under the supervised learning paradigm, which implies that they require abundant labeled data

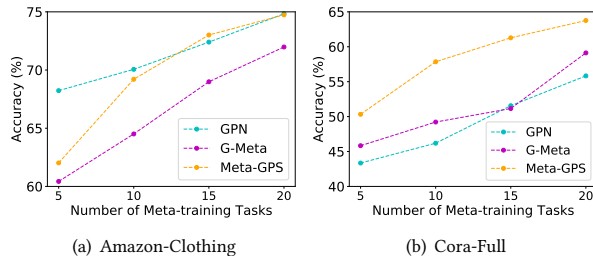

(a) Amazon-Clothing      (b) Cora-Full

**Figure 1: Model performance varies with the number of meta-training tasks across different datasets.**

to achieve satisfactory classification performance [5, 43]. Given the limited number of labeled nodes per class, known as few-shot cases [25, 62], these models suffer from severe overfitting, leading to a significant performance decline [15, 21].

Meta-learning has emerged as a viable option for effectively learning from limited labeled data. Its core concept is to train on tasks instead of instances as training units, aiming to capture the differences between tasks to enhance the model generalizability [14]. Several pioneering models [50, 65] have attempted to leverage integrate GNNs and meta-learning techniques to address graph few-shot learning problems. However, these graph meta-learning models all assume the existence of abundant accessible meta-training tasks to extract generalizable meta-knowledge for rapid adaptation to meta-testing tasks with only a few labeled instances. In other words, their outstanding performance critically depends on a wide range of meta-training tasks. For many real-world applications, due to the difficulty of task generation or data collection, we may not be able to obtain an adequate number of meta-training tasks [19, 43, 60]. For molecular property prediction, labeling newly discovered chemical compounds requires extensive domain knowledge and expensive wet-lab experiments [11]. Moreover, even after annotation, the currently known chemical properties (*i.e.*, classes) are limited, encompassing only common molecular characteristics such as polarity, solubility, and toxicity [26].

To further support our argument, we select three representative graph meta-learning models (*i.e.*, GPN [5], G-Meta [15], and Meta-GPS [25]) and evaluate their performance under varying numbers of meta-training tasks in Fig. 1. We distinctly observe that as the number of available meta-training tasks decreases, the overall performance of all methods greatly deteriorates. Because they tend to memorize meta-training tasks directly, which significantly constrains their generalization ability to novel tasks in the meta-testing stage [36]. This naturally raises a pressing question for us in more realistic scenarios: *How can we perform graph few-shot learning in scenarios with fewer tasks to extract as much transferable meta-knowledge as possible, thereby enhancing the model generalization performance?* Regarding this, although some recent studies [17, 42] have made some efforts on this issue, they primarily employ

intricate network architectures to endow models with favorable characteristics, yet there is still room for improvement. We argue that there are two serious issues for our focused scenarios, which greatly hamper the model performance. *On the one hand*, there are only limited support samples available for training within each meta-training task, which complicates the accurate reflection of the real data distribution. Therefore, it poses a challenge for the model to effectively capture the data characteristics, severely affecting its inductive bias capability [33]. *On the other hand*, when there are only limited meta-training tasks available, the model tends to directly fit the biased task distribution. This implies not only a shortage of data for each task, but also a reduced number of available tasks. The combined effect of these two factors increases the unnecessary oscillation during predictions outside the training examples, leading to reduced generalization capability.

To address these issues mentioned above, we develop a **SiM**ple yet effect**I**ve approach for graph few-shot **L**earning with f**E**wer tasks, namely, **SMILE**. Specifically, given the graph data, we first obtain discriminative node embeddings using our designed graph encoder. In this process, we introduce node degrees as prior information to fully utilize valuable information present in the existing graph. Then, we introduce a dual-level mixup strategy that operates on the obtained hidden node representations, consisting of both within-task and across-task mixup. The former involves random sampling of two instances from the same category within a task and applies linear interpolation to generate new samples, thereby enriching the data distribution. The latter requires computing class prototypes in two randomly selected original meta-training tasks, and then linearly interpolating class prototypes from different tasks to generate new tasks, thereby densifying the task distribution. These two employed strategies effectively mitigate the adverse effects caused by sample and task scarcity. Empirically, despite its simplicity and the absence of sophisticated techniques, the proposed approach demonstrates remarkable performance. Furthermore, we provide a theoretical elucidation of the underlying mechanism of our method, demonstrating its ability to constrict the upper bound of generalization error and consequently achieve superior generalization. In summary, our contributions can be summarized as follows:

- We propose a simple yet effective approach, SMILE, which leverages dual-level task mixup technique and incorporates the node degrees prior information, for graph few-shot learning with fewer tasks.
- We theoretically analyze the reasons why our approach works, demonstrating its ability to enhance generalization performance by regularizing model weights.
- We conduct extensive experiments on the several datasets, and the results show that SMILE can considerably outperform other competitive baselines by a large margin with in-domain and cross-domain settings.

## 2 Related Work

**Few-shot Learning.** Few-shot learning aims to quickly adapt meta-knowledge acquired from previous tasks to novel tasks with only a small number of labeled samples, thereby enabling few-shot generalization of machine learning algorithms [6, 7, 14]. Typically,

there are three main strategies to solve few-shot learning. Some methods [24, 29, 40, 41, 48] utilize prior knowledge to constrain the hypothesis space at the *model level*, learning a reliable model within the resulting smaller hypothesis space. A series of methods [8–10, 34, 38] improve the search strategy at the *algorithm level* by providing good initialization or guiding the search steps. Another line of works [13, 39, 57, 58, 66] augment tasks at the *data level* to obtain precise empirical risk minimizers. For the few-shot learning with limited tasks, there are various explorations in Euclidean data, such as images and texts. For example, TAML [16] and MR-MAML [61] directly apply regularization on the few-shot learner to reduce their reliance on the number of tasks. Meta-aug [36] and MetaMix [58] perform data augmentation on individual tasks to enrich the data distribution. MLTI [60] and Meta-Inter [19] directly generate source tasks to densify the task distribution. Our approach differs from the aforementioned methods in that we simultaneously perform within-task and across-task interpolation, with each strategy playing a crucial role. Meanwhile, we integrated task prototypes into the mixup process and explicitly adopt those previously overlooked original tasks, resulting in superior performance, which can be supported by the results in **Appendices** F.3 and F.4. Moreover, previous methods are not applicable to graph-structured data, whereas SMILE introduces a strategy that leverages the prior information provided by the graph.

**Graph Few-shot Learning.** Inspired by the success of few-shot learning in computer vision [2, 20, 45] and natural language processing [1, 31, 49], few-shot learning on graphs has recently seen significant development [25, 42, 50, 51, 59]. The core concept of current mainstream methods is to develop complicated algorithms to address the problem of few-shot learning on graphs. For instance, Meta-GNN [65], G-Meta [15], and Meta-GPS [25] are all subjected to specific modifications based on the MAML [6] algorithm, employing a bi-level optimization strategy to learn better parameter initialization. While the above models yield fruitful results, their reliance on substantial and diverse of meta-training tasks, coupled with their high complexity, has impeded their further advancement. Recently, TLP [43] and TEG [17] attempt to alleviate the limited diversity in meta-training datasets by using graph contrastive learning and equivariant neural networks, respectively. With the aid of sophisticated network designs, these methods have yielded promising results in graph few-shot scenarios. However, there is little effort to address the graph few-shot learning problem from the perspective of data augmentation.

## 3 Preliminary

Given a graph $\mathcal{G} = \{\mathcal{V}, \mathcal{E}, Z, A\}$, $\mathcal{V}$ and $\mathcal{E}$ represent the sets of nodes and edges, respectively. $Z \in \mathbb{R}^{n \times d}$ is the feature matrix of nodes and $A \in \mathbb{R}^{n \times n}$ is the corresponding adjacency matrix. Our model adheres to the prevalent meta-learning training paradigm, which involves training on sampled tasks. In this work, we mainly focus on few-shot node classification, which is the most prevailing and representative task in graph few-shot learning. Moreover, we highlight that in our focused scenarios, the number of available meta-training tasks sampled from an unknown task distribution is extremely small compared to traditional experimental settings, referred to as *few-shot node classification with fewer tasks*. Our goal

is to enable the model to effectively extract meta-knowledge even from such limited tasks, which can generalize to novel tasks in the meta-testing phase. For better understanding, we summarize the main symbols of this work in **Appendix** A.

## 4 Method

In this section, we will detail our proposed SMILE, which consists of two components: node representation learning and dual-level mixup strategy. To facilitate better understanding, we present the overall framework of the model in Fig. 2.

### 4.1 Node Representation Learning

Generally, the initial step involves encoding the nodes within the graph into a latent space, thereby transforming them into low-dimensional hidden vectors. GNNs have become the foremost choice for node embedding due to its powerful representational capabilities on graphs. It follows a message-passing mechanism, continuously aggregating messages from neighboring nodes to iteratively update the embedding of the target node. Guided by the simple philosophy, we adopt the SGC model [55] to learn node embeddings. Specifically, which can be defined as:

$$H = \check{A} \cdots \check{A} Z W^{(0)} W^{(1)} \cdots W^{(\ell-1)} = \check{A}^\ell Z W^*, \quad (1)$$

where $\check{A} = \hat{D}^{-1/2} \hat{A} \hat{D}^{-1/2}$ is the symmetric normalized adjacency matrix with added self-loops, *i.e.*, $\hat{A} = A + I$. $\hat{D}_i = \sum_j \hat{A}_{i,j}$ denotes the corresponding degree matrix. $W^*$ is the collapsing weight matrices. After performing graph convolution operations, we can obtain the node vectors $H \in \mathbb{R}^{n \times d}$ that simultaneously encode node features and topology structure.

Given that few-shot models are highly noise-sensitivity [63], it is necessary to incorporate more prior knowledge to refine representations. Such prior knowledge is often reflected on node degree about the node popularity and importance [35]. Therefore, we consider explicitly incorporating it to evaluate each node. Specifically, we first adopt another SGC to derive the interaction weights $\kappa \in \mathbb{R}^{n \times 1}$ for all nodes. Then, based on the node degree information, we obtain the node centralities $\alpha \in \mathbb{R}^{n \times 1}$ to perform degree normalization for adjusting $\kappa$. Finally, we acquire the refined node representations $X \in \mathbb{R}^{n \times d}$ using the adjusted scores $\beta \in \mathbb{R}^{n \times 1}$. The above procedures can be formulated as follows:

$$\begin{aligned} \kappa &= \check{A}^\ell Z W, \quad \alpha = \log(\{\hat{D}_i\}_{i=1}^n), \\ \beta &= \text{softmax}(\delta(\alpha \odot \kappa)), \quad X = \beta \odot H, \end{aligned} \quad (2)$$

where $W \in \mathbb{R}^{d \times 1}$ is the trainable parameters and $\delta(\cdot)$ is the sigmoid function. $\odot$ denotes the element-wise product.

After completing the node representation learning, we introduce the few-shot setting by defining some key notations. The meta-training tasks $\mathcal{D}_{org} = \{\mathcal{T}_t\}_{t=1}^{T_{org}}$ are sampled from a task distribution $p(\mathcal{T})$, where each task contains a support set $\mathcal{S}_t = \{(X_{t,i}^s, Y_{t,i}^s)\}_{i=1}^{n_s}$ and a query set $Q_t = \{(X_{t,i}^q, Y_{t,i}^q)\}_{i=1}^{n_q}$, denoted as $\mathcal{T}_t = \{\mathcal{S}_t, Q_t\}$. Here, $X_{t,i}^*$ and $Y_{t,i}^* \in \mathcal{Y}_{tra}$ denote the node embeddings and its label, where $\mathcal{Y}_{tra}$ denotes the set of base classes. For the meta-testing task $\mathcal{T}_{tes} = \{\mathcal{S}_{tes}, Q_{tes}\} = \{\{(X_{tes,i}^s, Y_{tes,i}^s)\}_{i=1}^{n_s}, \{(X_{tes,i}^q, Y_{tes,i}^q)\}_{i=1}^{n_q}\}$, it is composed in the same way as the meta-training task $\mathcal{T}_t$, with the only difference being that the node label belong to the novel

class set $\mathcal{Y}_{tes}$, which is disjoint from $\mathcal{Y}_{tra}$, *i.e.*, $\mathcal{Y}_{tra} \cap \mathcal{Y}_{tes} = \emptyset$. When the support set $\mathcal{S}_{tes}$ consists of $N$ sampled classes, each with $K$ nodes, we refer to it as an $N$-way $K$-shot problem. The construction of $Q_{tes}$ is the same as $\mathcal{S}_{tes}$, except that each class has $M$ nodes. Typically, the model is first trained on the meta-training tasks $\mathcal{D}_{org}$. During the meta-testing stage, the model is fine-tuned on $\mathcal{S}_{tes}$ and then is evaluated the performance on $Q_{tes}$.

### 4.2 Dual-level Mixup Strategy

If we directly conduct few-shot learning on the refined representations, the model's performance would be degraded due to overfitting and constrained generalization. Therefore, we introduce a dual-level mixup strategy, including within-task and across-task mixup, to deal with this issue. Next, we will provide detailed descriptions of each technique.

*4.2.1 Within-task Mixup.* Due to the exceedingly restricted number of sampled nodes in both the support set and query set for each task during the meta-training phase, the efficiency of the meta-training is considerably compromised. Hence, we propose using the within-task mixup strategy to generate more samples for increasing the diversity of the data. Concretely, for a given meta-training task $\mathcal{T}_t$, we perform random sampling on the support set $\mathcal{S}_t$ and query set $Q_t$, selecting two samples $i$ and $j$ from the same category $k$ for linear interpolation to generate a new one $r$. The above procedure can be formulated as:

$$X_{t,r;k}^{\prime s} = \lambda X_{t,i;k}^s + (1-\lambda) X_{t,j;k}^s, \quad X_{t,r;k}^{\prime q} = \lambda X_{t,i;k}^q + (1-\lambda) X_{t,j;k}^q, \quad (3)$$

where $\lambda \in [0, 1]$ is sampled from the Beta distribution $Beta(\eta, \gamma)$ specified by $\eta$ and $\gamma$.

Here, we do not perform label interpolation as the labels of the two sampled nodes are the same, resulting in identical labels for the generated node. There are two reasons for this. First, interpolating samples from different categories would make it difficult to compute prototypes of the mixed labels while expanding the node label space of the original task. Second, this would pose intricate troubles for the subsequent across-task interpolation.

We iteratively apply Eq.3 to generate the additional support set $\mathcal{S}_t' = \{(X_{t,i}^{\prime s}, Y_{t,i}^s)\}_{i=1}^{n_{s'}}$ and query set $Q_t' = \{(X_{t,i}^{\prime q}, Y_{t,i}^q)\}_{i=1}^{n_{q'}}$, which are subsequently merged with the original corresponding sets to obtain the augmented task $\mathcal{T}_t$ (To avoid introducing extra symbols, we consistently use $\mathcal{T}_t$ to denote the task that undergoes within-task mixup in the following sections.), *i.e.*, $\mathcal{T}_t = \{\mathcal{S}_t \cup \mathcal{S}_t', Q_t \cup Q_t'\}$. The number of nodes in the amplified support and query sets of the augmented task $\mathcal{T}_t$ are $m' = n_s + n_{s'}$ and $m = n_q + n_{q'}$, respectively.

*4.2.2 Across-task Mixup.* Solely conducting within-task mixup does not address the issue of the limited number of tasks. Therefore, we utilize across-task mixup to directly create new tasks, densifying the task distribution. Specifically, *in the first step*, we randomly select two tasks, $\mathcal{T}_i$ and $\mathcal{T}_j$, from the given meta-training tasks $\mathcal{D}_{org} = \{\mathcal{T}_t\}_{t=1}^{T_{org}}$. *In the second step*, we randomly sample a class $k$ from the support set $\mathcal{S}_i$ of $\mathcal{T}_i$ and a class $k'$ from the support set $\mathcal{S}_j$ of $\mathcal{T}_j$, and then compute class-specified support prototypes. This

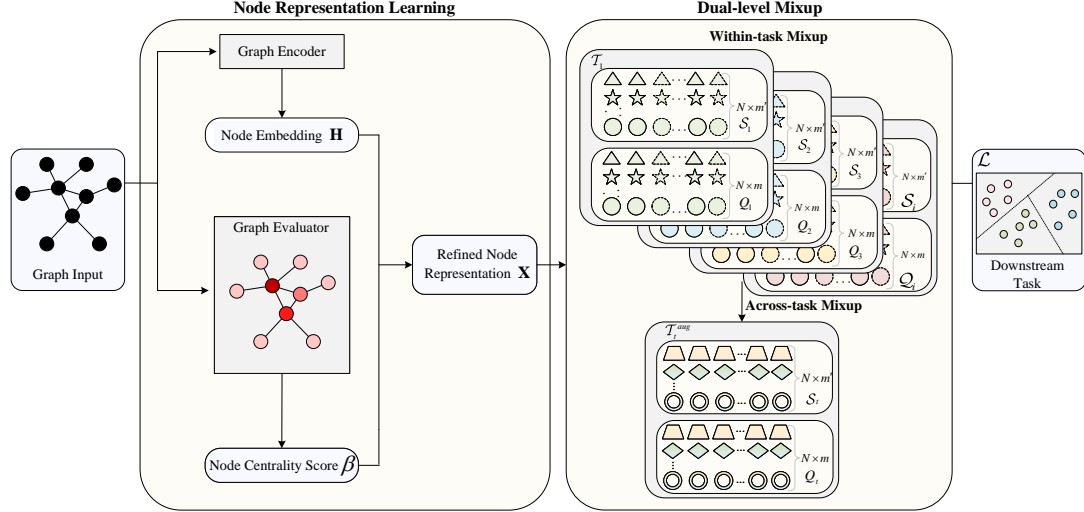

**Figure 2: The overall architecture of SMILE.**

procedure can be expressed as:

$$
C_{i;k}^s = \frac{1}{|\mathcal{S}_{i;k}|} \sum_{(X_{i,\varrho}^s, Y_{i,\varrho}^s) \in \mathcal{S}_i} \mathbb{I}_{Y_{i,\varrho}=k} X_{i,\varrho}^s,
$$

$$
C_{j;k'}^s = \frac{1}{|\mathcal{S}_{j;k'}|} \sum_{(X_{j,\varrho}^s, Y_{j,\varrho}^s) \in \mathcal{S}_j} \mathbb{I}_{Y_{j,\varrho}=k'} X_{j,\varrho}^s, \qquad (4)
$$

where $\mathbb{I}(\cdot)$ is the indicator function that is 1 when $Y_{i,\varrho} = k$, and 0 otherwise. Similarly, we can obtain query prototypes $C_{i;k}^q$ for class $k$ and $C_{j;k'}^q$ for class $k'$ by applying Eq.4 to the query sets $Q_i$ of $\mathcal{T}_i$ and $Q_j$ of $\mathcal{T}_j$.

*In the third step*, we individually perform feature-level linear interpolation on the support prototypes and query prototypes to generate new samples. Considering that different tasks have different label spaces, we directly treat the label associated with the interpolated data as a new class $\tilde{k}$. We can formulate the above process as:

$$
\tilde{X}_{t,\varrho;\tilde{k}}^s = \lambda C_{i;k}^s + (1-\lambda) C_{j;k'}^s, \qquad \tilde{Y}_{t,\varrho;\tilde{k}}^s = \Phi(Y_{i;k}^s, Y_{j;k'}^s),
$$

$$
\tilde{X}_{t,\varrho;\tilde{k}}^q = \lambda C_{i;k}^q + (1-\lambda) C_{j;k'}^q, \qquad \tilde{Y}_{t,\varrho;\tilde{k}}^q = \Phi(Y_{i;k}^q, Y_{j;k'}^q), \qquad (5)
$$

where $\Phi(\cdot, \cdot)$ represents the label uniquely determined by the pair $(\cdot, \cdot)$. We perform $m'$ iterations for the support data and $m$ iterations for the query data in Eq.5, *i.e.*, $\{\tilde{X}_{t,\varrho;\tilde{k}}^s, \tilde{Y}_{t,\varrho;\tilde{k}}^s\}_{\varrho=1}^{m'}$ and $\{\tilde{X}_{t,\varrho;\tilde{k}}^q, \tilde{Y}_{t,\varrho;\tilde{k}}^q\}_{\varrho=1}^m$. Note that the sampled $\lambda$ each time is different.

*Finally*, we repeat the second and third steps $N$ times to construct an $N$-way $m'$-shot interpolation task $\mathcal{T}_t^{aug} = \{\tilde{\mathcal{S}}_t, \tilde{Q}_t\} = \{\{\tilde{X}_{t;\tilde{k}}^s, \tilde{Y}_{t;\tilde{k}}^s\}_{\tilde{k}=1}^N, \{\tilde{X}_{t;\tilde{k}}^q, \tilde{Y}_{t;\tilde{k}}^q\}_{\tilde{k}=1}^N\}$. We can conduct the above process multiple times to obtain the interpolated tasks $\mathcal{D}_{aug} = \{\mathcal{T}_t^{aug}\}_{t=1}^{T_{aug}}$ and merge them with the original tasks $\mathcal{D}_{org}$ to form the final meta-training tasks $\mathcal{D}_{all} = \mathcal{D}_{org} \cup \mathcal{D}_{aug}$. The number of tasks in $\mathcal{D}_{all}$ is $T = T_{org} + T_{aug}$.

**Model Training.** After performing the dual-level mixup operation, we adopt a classic metric-based episodic training for few-shot node classification. We first derive the prototype $C_k$ in the support set $\mathcal{S}_t$ of each task $\mathcal{T}_t$ from $\mathcal{D}_{all}$ with the manner shown in Eq.4.

Next, we optimize the parameters of the model by performing distance-based cross-entropy loss function on all query sets in $\mathcal{D}_{all}$ as:

$$
\mathcal{L} = \sum_{t=1}^{T} \sum_{i=1}^{m} \mathbb{I}_{Y_{t,i}=k} \log \frac{\exp(-d(\theta^\top X_{t,i}^q, C_k))}{\sum_{k'} \exp(-d(\theta^\top X_{t,i}^q, C_{k'}))}, \qquad (6)
$$

where $d(\cdot, \cdot)$ is the Euclidean distance function and $\theta$ is the trainable vector.

In the meta-testing stage, we do not perform any mixup operations for the evaluated task $\mathcal{T}_{tes}$. Actually, we first use the well-trained model to compute class prototypes on the support set, and then assign samples in the query set to their nearest prototype, defined as:

$$
C_k = \frac{1}{|\mathcal{S}_{tes,k}|} \sum_{(X_{tes,i}^s, Y_{tes,i}^s) \in \mathcal{S}_{tes}} \mathbb{I}_{Y_{tes,i}=k} X_{tes,i}^s,
$$

$$
Y_{tes,*}^q = \arg\min_k d(\theta^\top X, C_k). \qquad (7)
$$

We present the process of proposed SMILE in Algorithm 1. The time complexity analysis of SMILE are presented in **Appendix** B.

## 5 Theoretical Analysis

In this section, we theoretically analyze why our proposed SMILE, equipped with intra-task and inter-task mixup, can alleviate overfitting and exhibit better generalization capabilities. We first present the obtained key points: *SMILE can regularize the weight parameters in a distribution-dependent manner and reduce the upper bound of the generalization gap by controlling the Rademacher complexity.* Next, we elaborate on the proposed theorems to support the aforementioned points. For simplicity, we conduct a detailed theoretical analysis of SMILE in the binary classification scenario, assuming

**Algorithm 1** The process of SMILE

---

**Input:** A graph $\mathcal{G} = \{\mathcal{V}, \mathcal{E}, Z, A\}$.
**Output:** The well-trained SMILE.
1: // *Meta-training process*
2: **while** *not convergence* **do**
3:    Learn node embeddings using Eq.1.
4:    Refine node embeddings using Eq.2.
5:    Construct meta-training tasks $\mathcal{D}_{org}$.
6:    Perform within-task mixup to obtain the augmented task $\mathcal{T}_t$ using Eq.3.
7:    Perform across-task mixup to obtain the interpolated task $\mathcal{T}_t^{aug}$ using Eqs.4 and 5.
8:    Form the interpolated tasks $\mathcal{D}_{aug}$.
9:    Obtain the enriched meta-training tasks $\mathcal{D}_{all}$.
10:    Compute the prototypes of support set for each task using Eq.4.
11:    Optimize the model using Eq.6.
12: **end while**
13: // *Meta-testing process*
14: Construct meta-testing task $\mathcal{T}_{tes}$.
15: Compute the prototypes in $\mathcal{S}_{tes}$ using Eq.7.
16: Predict the node labels in $Q_{tes}$.

---

the use of preprocessed centralized dataset that satisfies the condition $\mathbb{E}_X[X] = 0$. Moreover, the proposed SMILE can be modeled as $f_\theta(Z) = \theta^\top g_\zeta(Z) = \theta^\top X$, where $g_\zeta(\cdot)$ denotes the graph encoder parameterized by $\zeta$. We consider using the loss from Eq.6 for tasks in $\mathcal{D}_{all}$. Particularly, the empirical loss function based on enriched training samples for binary classification can be simplified as:

$$\mathcal{L}(\mathcal{D}_{all}; \theta) = \sum_{t=1}^{T} \sum_{i=1}^{m} (1 + \exp(\langle X_{t,i}^q - (C_1 + C_2)/2, \theta \rangle))^{-1},$$

$$C_k = \frac{1}{|\mathcal{S}_{t,k}|} \sum_{(X_{t,i}^s, Y_{t,i}^s) \in \mathcal{S}_t} \mathbb{I}_{Y_{t,i} = k} X_{t,i}^s, \tag{8}$$

where $\langle \cdot, \cdot \rangle$ denote the dot product operation. The approximation of the loss function $\mathcal{L}(\mathcal{D}_{all}; \theta)$ in Eq.8 is formalized as:

$$\mathcal{L}(\mathcal{D}_{all}; \theta) \approx \mathcal{L}(\mathcal{D}_{org}; \theta) + \mathcal{L}(\bar{\lambda} \mathcal{D}_{org}; \theta) + \mathcal{M}(\theta), \tag{9}$$

where $\bar{\lambda} = \mathbb{E}_{\rho_\lambda}[\lambda]$ and $\mathcal{M}(\theta)$ is a quadratic regularization term with respect to $\theta$, defined as:

$$\mathcal{M}(\theta) = \mathbb{E}_{\mathcal{T}_t \sim p(\mathcal{T})} \mathbb{E}_{(X_t, Y_t) \sim q(\mathcal{T}_t)} \frac{\phi(P_t)(\phi(P_t) - 0.5)}{2(1 + \exp(P_t))} (\theta^\top \Sigma_X \theta) \tag{10}$$

in which $P_t = \langle X_t^q - (C_1 + C_2)/2, \theta \rangle$, $\phi(P_t) = \exp(P_t)/(1 + \exp(P_t))$, and $\Sigma_X = \mathbb{E}[XX^\top] = \frac{1}{m} \sum_{i=1}^{m} X_i X_i^\top$. The detailed proofs for Eqs.9 and 10 can be found in Lemma C.1 of Appendix C.

Eq.9 shows that SMILE imposes an additional regularization term on the trainable weights to constrain the solution space, thereby reducing the likelihood of overfitting.

To define the generalization gap problem formally, we introduce a function class of the dual form related to the regularization term in Eq.9, as shown in Eq.11.

$$\mathcal{F}_\nu = \{X \to \theta^\top X : \theta^\top \Sigma_X \theta \le \nu\}. \tag{11}$$

Moreover, we represent the expected risk R and empirical risk $\hat{R}$ as follows:

$$R = \mathbb{E}_{\mathcal{T}_i \sim p(\mathcal{T})} \mathbb{E}_{(X_j, Y_j) \sim q(\mathcal{T}_i)} \mathcal{L}(f_\theta(X_j), Y_j),$$

$$\hat{R} = \mathbb{E}_{\mathcal{T}_i \sim \hat{p}(\mathcal{T})} \mathbb{E}_{(X_j, Y_j) \sim \hat{q}(\mathcal{T}_i)} \mathcal{L}(f_\theta(X_j), Y_j). \tag{12}$$

Then, we present the following theorem for improved generalization gap brought by SMILE.

**Theorem 5.1.** *Assume that* X, Y *and* $\theta$ *are bounded. For all* $f \in \mathcal{F}_\nu$, *where* $\theta$ *satisfies* $\theta^\top \Sigma_X \theta \le \nu$, *we have the following generalization gap bound, with probability at least* $(1 - \epsilon)$ *over the training samples,*

$$|\hat{R} - R| \le 2 \left( \sqrt{\frac{\nu \cdot rank(\Sigma_X)}{m}} + \sqrt{\frac{\nu}{T}} \left( rank(\Sigma_X) \right) \right)$$

$$+ 3 \left( \sqrt{\frac{\log(2/\epsilon)}{2m}} + \sqrt{\frac{\log(2/\epsilon)}{2T}} \right), \tag{13}$$

*where* $m$ *and* T *denote the number of nodes in the query set and the number of meta-training tasks.*

Based on Theorem 5.1, we can obtain several in-depth findings. On the one hand, SMILE induces a regularized weight space for $\theta$, leading to a smaller $\nu$. On the other hand, the introduced intra-task and inter-task interpolations increase $m$ and T simultaneously. These two aspects work together to reduce the upper bound of the generalization gap of SMILE and alleviate overfitting.

According to the above theorem, we can naturally confirm the following corollary.

**Corollary 5.2.** *Let* $|\hat{R} - R|$ *and* $|\hat{R}_{ori} - R_{ori}|$ *denote the model generalization bounds trained under our proposed task augmentation strategy and standard training strategy, respectively. We have the following inequality holding,*

$$U(|\hat{R} - R|) \le U(|\hat{R}_{ori} - R_{ori}|), \tag{14}$$

*where* $U(\cdot)$ *denotes the operation of taking the upper bound.*

Corollary 5.2 suggests that SMILE achieves tight generalization bound than other models trained in a standard way.

Suppose the empirical distribution of source tasks in the meta-training be $\hat{\mathbb{P}}$ and the expected distribution of the query set of target tasks be $\mathbb{Q}$, then during the adaptation process, we expect to reduce the data-dependent upper bound, defined as $\sup_{f \in \mathcal{F}} |\mathbb{E}_{\hat{\mathbb{P}}} - \mathbb{E}_{\mathbb{Q}}|$. Empirically, when source tasks and target tasks are more similar, the model is more likely to extract generalizable meta-knowledge from source tasks to quickly adapt to target tasks. Theoretically, we present the Theorem 5.3, which demonstrates the reduction of the upper bound between $\hat{\mathbb{P}}$ and $\mathbb{Q}$ induced by our proposed strategy.

**Theorem 5.3.** *Assume the source tasks and target tasks are drawn from distribution* $\hat{\mathbb{P}}$ *and* $\mathbb{Q}$, *and they are independent. For* $\epsilon > 0$, *with probability at least* $(1 - \epsilon)$ *over the draws of samples, we have the following upper bound between data distributions,*

$$\sup_{f \in \mathcal{F}} |\mathbb{E}_{\hat{\mathbb{P}}} - \mathbb{E}_{\mathbb{Q}}| \le \left( 2\sqrt{\nu \cdot rank(\Sigma_X)} + \sqrt{\frac{\log(1/\epsilon)}{2}} \right) \left( \sqrt{\frac{1}{m}} + \sqrt{\frac{1}{n_q}} \right), \tag{15}$$

*where* $n_q$ *denotes the number of nodes in the query set of the meta-testing task.*

We can draw the conclusion that the introduction of intra-task interpolation leads to an increase in the value of $m$. Additionally, according to Eq.9, the regularization effect can result in a decrease of $\nu$. Consequently, Theorem 5.3 suggests that our method has the capability to diminish the disparity between the distributions of the source task and target task, facilitating the extraction of pertinent knowledge and, in turn, enhancing the model's generalization. All detailed proofs can be found in **Appendix** C.

## 6  Experiment

**Datasets.** To demonstrate the effectiveness of our approach, we conduct few-shot node classification with fewer tasks on four selected prevalent datasets widely used by previous researches, including **Amazon-Clothing** [28], **CoraFull** [3], **Amazon-Electronics** [28], and **DBLP** [44]. Table 1 shows the statistics of these datasets. Concisely, # Nodes and # Edges represent the number of nodes and edges in the dataset, respectively. # Features denotes the dimension of the initialized node features, and # Labels is the number of classes. Class Splits represents the number of classes used for meta-training/meta-validation/meta-testing. The detailed descriptions of these evaluated datasets can be found in **Appendix** D.

**Table 1: Statistics of the datasets.**

| Dataset | # Nodes | # Edges | # Features | # Labels | Class Splits |
|---|---|---|---|---|---|
| Amazon-Clothing | 24,919 | 91,680 | 9,034 | 77 | 40/17/20 |
| Cora-Full | 19,793 | 65,311 | 8,710 | 70 | 25/20/25 |
| Amazon-Electronics | 42,318 | 43,556 | 8,669 | 167 | 90/37/40 |
| DBLP | 40,672 | 288,270 | 7,202 | 137 | 80/27/30 |

**Baselines.** We mainly select three types of baselines for comparison to verify the superiority of the proposed SMILE. *Traditional meta-learning* methods consist of **Protonet** [40] and **MAML** [6]. *Meta-learning with fewer tasks* methods comprise **MetaMix** [58], **MLTI** [60], and **Meta-Inter** [19]. *Graph meta-learning* methods include **Meta-GNN** [65], **GPN** [5], **G-Meta** [15], **Meta-GPS** [25], **X-FNC** [51], **COSMIC** [53], **TLP** [43], and **TEG** [17]. We provide the descriptions and implementations of these baselines in **Appendix** E.

**Implementation Details.** In the *node representation learning* stage, we adopt 2-layer SGC with 16 hidden units. In the *dual-level mixup* stage, we uniformly set the two parameters involved in the beta distribution to 0.5, *i.e.*, $\eta = \gamma = 0.5$. Moreover, in within-task mixup, we generate the same number of nodes as the original support and query set in each meta-training task by default, that is, $n_{s'} = n_s$, $n_{q'} = n_q$. In across-task mixup, we generate as many interpolated tasks as original tasks, that is $T_{aug} = T_{org}$. In the cross-domain setting, we meta-train the model on one source domain and then meta-test it on another target domain. To ensure fair comparison, we perform meta-training on the same sampled tasks for all models. Moreover, we evaluate the performance of our model using the average accuracy (Acc) and macro-F1 (F1) score across 50 randomly selected meta-testing tasks.

## 7  Result

**Model Performance.** We present the results of our proposed SMILE and other models under both *in-domain* and *cross-domain*

settings with different number of tasks across several datasets in Tables 2 and 3. According to above results, we can obtain the following in-depth analysis. We can find that our approach achieves the best performance across varying numbers of meta-training tasks in both in-domain and cross-domain settings for all datasets, demonstrating its superiority in dealing with graph few-shot learning with fewer tasks. One plausible reason is that we explicitly introduce degree-based prior in the node representation stage, resulting in more discriminative features beneficial for subsequent tasks. Furthermore, we employ a dual-level mixup strategy, enriching the diversity of both within-task and across-task data, effectively alleviating the negative impact of data and task scarcity. These strategies facilitate the model to extract more transferable meta-knowledge, thereby greatly enhancing its generalization capability.

We find that graph meta-learning models represented by COSMIC and TEG perform well in scenarios with more tasks across multiple datasets, which aligns with our expectations. These models are specifically designed for graph few-shot learning and utilize unique few-shot algorithms that enable them to achieve discriminative node representations with limited labeled data. However, they struggle in scenarios with fewer tasks, performing significantly worse than our model. This is because they can only extract sufficient transferable meta-knowledge when there are ample meta-training tasks. Moreover, the meta-learning with fewer tasks models equipped with SGC, such as MLTI and Meta-Inter, also demonstrate impressive performance in the both in-domain and cross-domain experimental settings. We attribute this phenomenon to the specific strategies employed by these models to mitigate the negative effects caused by limited tasks. However, this type of models still significantly lag behind our model, as they do not incorporate degree-based prior knowledge and fail to address scarcity issues from both data and task perspectives simultaneously. Additionally, traditional meta-learning methods consistently underperform compared to other methods because they completely overlook the important structural information in the graph.

Also, we show additional results of these models under different experimental settings in the **Appendix** F, including model performance with sufficient tasks and across various graph encoders and so on, due to the space constraints.

**Ablation Study.** To demonstrate the effectiveness of our adopted strategies, we design several model variants. (I) *vanilla mixup*: Without the dual-level mixup, we first compute the prototypes using the support set, and then perform vanilla mixup to query set, which involves mixing the features and generates soft labels. (II) *internal mixup*: In the absence of dual-level mixup, we directly perform mixup on different classes within each task and generate corresponding hard labels, treating them as novel classes. (III) *w/o within and across*: We simultaneously discard both within-task and across-task mixup operations. (IV) *w/o within*: We delete the within-task operation and leave the across-task one. (V) *w/o across*: We remove the across-task strategy and retain the within-task one. (VI) *w/o degree*: We exclude the utilization of degree information and solely employ the vanilla SGC for node representation learning.

According to Table 4, it is evident that the employed strategies have a favorable impact on the model performance. The introduction of dual-level mixup enriches the samples within each task

**Table 2: Results (%) of different models using fewer tasks on datasets under the 5-way 5-shot *in-domain* setting. Bold: best (based on the pairwise t-test with 95% confidence). Underline: runner-up.**

| Model | Amazon-Clothing | | | | | | | | CoraFull | | | | | | | |
|---|---|---|---|---|---|---|---|---|---|---|---|---|---|---|---|---|
| | 5 tasks | | 10 tasks | | 15 tasks | | 20 tasks | | 5 tasks | | 10 tasks | | 15 tasks | | 20 tasks | |
| | Acc | F1 | Acc | F1 | Acc | F1 | Acc | F1 | Acc | F1 | Acc | F1 | Acc | F1 | Acc | F1 |
| Protonet | 49.17 | 48.36 | 53.51 | 52.55 | 55.82 | 54.99 | 57.99 | 57.14 | 37.20 | 35.98 | 40.14 | 38.89 | 43.90 | 42.96 | 45.58 | 44.34 |
| MAML | 44.90 | 43.66 | 45.67 | 44.44 | 46.29 | 44.97 | 46.90 | 45.60 | 38.15 | 36.83 | 42.26 | 41.28 | 44.21 | 43.95 | 46.37 | 45.43 |
| MetaMix | 78.32 | 78.22 | 78.66 | 78.52 | 80.16 | 79.15 | 81.09 | 80.52 | 62.95 | 62.25 | 64.20 | 63.95 | 65.72 | 64.19 | 67.59 | 66.26 |
| MLTI | 79.19 | 78.59 | 79.91 | 78.92 | 80.22 | 79.39 | 81.27 | 80.86 | 63.19 | 63.06 | 65.72 | 65.69 | 66.25 | 64.92 | 67.15 | 66.10 |
| Meta-Inter | 79.92 | 79.22 | 80.12 | 79.56 | 80.55 | 79.90 | 81.26 | 81.05 | 63.82 | 63.36 | 66.59 | 65.92 | 67.19 | 65.50 | 68.22 | 67.59 |
| Meta-GNN | 55.29 | 50.44 | 57.19 | 53.65 | 62.29 | 59.55 | 70.19 | 67.22 | 42.96 | 40.83 | 45.09 | 42.87 | 47.15 | 45.38 | 49.88 | 48.12 |
| GPN | 68.23 | 67.16 | 70.06 | 69.57 | 72.40 | 71.95 | 72.81 | 71.56 | 43.35 | 42.08 | 46.19 | 44.81 | 51.56 | 50.24 | 55.83 | 54.76 |
| G-Meta | 60.43 | 60.11 | 64.51 | 63.74 | 68.99 | 67.96 | 71.98 | 72.75 | 45.84 | 44.27 | 49.22 | 48.91 | 51.15 | 50.53 | 59.12 | 58.56 |
| Meta-GPS | 62.02 | 59.76 | 69.21 | 69.04 | 73.01 | 71.92 | 75.74 | 74.85 | 50.33 | 48.22 | 57.85 | 54.86 | 61.28 | 60.11 | 63.76 | 62.28 |
| X-FNC | 69.12 | 68.29 | 72.12 | 71.11 | 75.19 | 74.63 | 79.26 | 78.02 | 55.06 | 53.10 | 61.53 | 60.29 | 65.22 | 64.10 | 66.09 | 65.12 |
| COSMIC | 75.66 | 74.92 | 76.39 | 75.72 | 77.92 | 76.59 | 78.36 | 77.39 | 62.29 | 60.39 | 65.39 | 64.80 | 66.72 | 65.72 | 68.29 | 67.20 |
| TLP | 71.39 | 70.39 | 73.39 | 72.52 | 74.72 | 73.36 | 75.60 | 74.29 | 51.79 | 49.72 | 56.72 | 55.79 | 57.72 | 56.73 | 57.99 | 57.30 |
| TEG | 78.55 | 77.92 | 80.26 | 79.30 | 80.82 | 79.99 | 81.19 | 80.16 | 62.89 | 61.26 | 68.29 | 67.39 | 68.59 | 67.55 | 70.06 | 69.29 |
| SMILE | **82.80** | **82.49** | **83.46** | **82.88** | **83.92** | **83.33** | **84.66** | **84.52** | **66.34** | **65.70** | **71.72** | **71.15** | **70.78** | **70.19** | **72.60** | **72.10** |

| Model | Amazon-Electronics | | | | | | | | DBLP | | | | | | | |
|---|---|---|---|---|---|---|---|---|---|---|---|---|---|---|---|---|
| | 5 tasks | | 10 tasks | | 15 tasks | | 20 tasks | | 5 tasks | | 10 tasks | | 15 tasks | | 20 tasks | |
| | Acc | F1 | Acc | F1 | Acc | F1 | Acc | F1 | Acc | F1 | Acc | F1 | Acc | F1 | Acc | F1 |
| Protonet | 46.20 | 45.09 | 49.56 | 48.57 | 51.98 | 51.05 | 54.03 | 53.20 | 46.57 | 45.47 | 50.90 | 49.81 | 51.02 | 49.74 | 52.09 | 51.05 |
| MAML | 34.34 | 33.42 | 34.76 | 33.76 | 35.42 | 34.41 | 35.91 | 34.95 | 39.71 | 38.86 | 40.34 | 39.58 | 40.70 | 39.85 | 41.31 | 40.58 |
| MetaMix | 61.96 | 61.82 | 63.72 | 63.66 | 65.19 | 64.92 | 66.15 | 65.72 | 72.12 | 71.15 | 73.19 | 72.12 | 75.16 | 73.95 | 76.22 | 74.79 |
| MLTI | 62.25 | 62.02 | 65.26 | 65.09 | 66.72 | 65.59 | 67.19 | 66.22 | 72.36 | 71.96 | 72.92 | 72.55 | 73.22 | 73.10 | 75.10 | 74.95 |
| Meta-Inter | 62.79 | 62.56 | 65.76 | 65.52 | 67.19 | 66.15 | 68.99 | 67.29 | 72.52 | 72.11 | 73.19 | 72.99 | 74.28 | 73.25 | 75.29 | 75.10 |
| Meta-GNN | 40.52 | 39.74 | 46.16 | 45.87 | 48.92 | 47.93 | 50.86 | 50.07 | 50.68 | 49.04 | 53.86 | 49.67 | 59.72 | 59.36 | 65.49 | 62.12 |
| GPN | 49.08 | 47.91 | 51.12 | 49.98 | 54.24 | 53.23 | 56.69 | 55.62 | 70.26 | 69.13 | 74.42 | 73.48 | 76.02 | 75.03 | 76.61 | 75.60 |
| G-Meta | 43.29 | 42.20 | 49.57 | 52.90 | 56.96 | 56.58 | 60.41 | 59.91 | 53.08 | 48.13 | 55.92 | 53.64 | 57.82 | 56.76 | 63.17 | 62.85 |
| Meta-GPS | 46.11 | 43.62 | 57.90 | 56.20 | 67.73 | 66.69 | 70.13 | 69.15 | 56.59 | 54.12 | 65.20 | 63.20 | 73.00 | 72.35 | 75.16 | 73.19 |
| X-FNC | 59.26 | 56.39 | 63.72 | 62.10 | 69.82 | 67.63 | 71.36 | 70.02 | 69.06 | 68.10 | 72.53 | 71.29 | 74.29 | 73.22 | 76.19 | 75.20 |
| COSMIC | 64.06 | 63.02 | 67.36 | 66.32 | 68.22 | 67.09 | 70.16 | 69.30 | 71.29 | 70.19 | 72.09 | 70.80 | 73.02 | 71.20 | 75.16 | 72.22 |
| TLP | 63.09 | 62.19 | 64.30 | 63.59 | 65.72 | 64.32 | 67.18 | 66.72 | 71.26 | 70.75 | 72.87 | 72.09 | 73.39 | 73.06 | 75.16 | 74.69 |
| TEG | 65.90 | 64.62 | 67.29 | 66.22 | 69.80 | 68.29 | 72.12 | 71.16 | 72.59 | 72.26 | 73.79 | 72.19 | 75.52 | 74.50 | 76.26 | 75.12 |
| SMILE | **67.30** | **66.30** | **70.76** | **70.05** | **73.48** | **72.66** | **75.42** | **75.42** | **75.88** | **75.05** | **76.64** | **75.77** | **79.56** | **78.77** | **80.50** | **79.61** |

and provides diverse tasks, making significant contributions to enhancing the model. The adopted degree-based prior information also improves the model by learning expressive node embeddings, especially in cross-domain setting. When removing the degree information, the performance drastically declines. One plausible reason is that this module explicitly utilizes structural prior knowledge from the target graph domain, benefiting downstream tasks. Additionally, the results of model variants I and II demonstrate that performing label interpolation within each task, whether generating soft or hard labels, degrades the model performance. The possible reason is that the introduced mixing labels can confuse the model.

**Hyperparameter Sensitivity.** In the 5-way 5-shot in-domain setting, we investigate the impact of two primary hyperparameters on the model performance: the ratio of generated nodes to the original nodes per task (*i.e.*, $\frac{n'_s+n'_q}{n_s+n_q}$), and the number of generated tasks $T_{aug}$. Notably, when the studied hyperparameter changes, we set others to their default values. The results are presented in

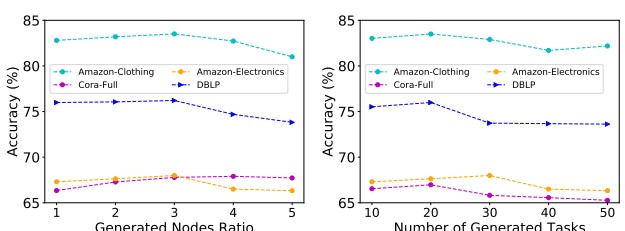

**Figure 3: Results vary with hyperparameters.**

Fig. 3. We can observe that both parameters demonstrate similar trends, with the model performance showing an initial increase followed by a decrease. We attribute this behavior to the substantial enrichment of data diversity by increasing the number of nodes within each task or the number of tasks. However, beyond a certain threshold, the introduced additional data fails to further densify the

**Table 3: Results (%) of different models using fewer tasks on datasets under the 5-way 5-shot *cross-domain* setting. A→B denotes the model is trained on A and evaluated on B.**

| Dataset | Amazon-Clothing→CoraFull | | | | | | | | CoraFull→Amazon-Clothing | | | | | | | |
| | 5 tasks | | 10 tasks | | 15 tasks | | 20 tasks | | 5 tasks | | 10 tasks | | 15 tasks | | 20 tasks | |
| | Acc | F1 | Acc | F1 | Acc | F1 | Acc | F1 | Acc | F1 | Acc | F1 | Acc | F1 | Acc | F1 |
|---|---|---|---|---|---|---|---|---|---|---|---|---|---|---|---|---|
| Protonet | 20.72 | 7.90 | 22.84 | 10.89 | 29.70 | 15.91 | 32.96 | 18.19 | 24.84 | 13.18 | 29.96 | 22.49 | 32.84 | 26.01 | 34.58 | 29.90 |
| MAML | 20.40 | 13.51 | 20.74 | 13.19 | 26.68 | 12.22 | 30.19 | 15.92 | 23.56 | 12.10 | 27.35 | 20.16 | 30.19 | 23.95 | 32.96 | 27.96 |
| MetaMix | 31.96 | 28.76 | 33.12 | 31.22 | 35.22 | 33.19 | 37.15 | 35.55 | 34.76 | 31.66 | 36.25 | 33.69 | 39.72 | 37.16 | 41.26 | 39.25 |
| MLTI | 33.29 | 30.12 | 35.16 | 32.29 | 38.25 | 35.52 | 40.22 | 38.29 | 35.12 | 33.49 | 37.22 | 35.29 | 42.19 | 41.52 | 45.66 | 43.95 |
| Meta-Inter | 34.72 | 32.19 | 35.76 | 34.26 | 40.16 | 37.59 | 42.29 | 40.32 | 41.76 | 39.59 | 43.22 | 41.57 | 44.11 | 42.25 | 47.29 | 45.55 |
| Meta-GNN | 26.36 | 20.99 | 30.50 | 26.72 | 33.22 | 30.15 | 35.99 | 32.16 | 32.16 | 22.39 | 35.22 | 26.62 | 38.16 | 29.35 | 39.66 | 32.90 |
| GPN | 35.86 | 34.81 | 39.38 | 38.03 | 41.10 | 39.82 | 41.96 | 41.15 | 40.08 | 38.73 | 41.78 | 40.67 | 43.90 | 42.87 | 45.04 | 44.30 |
| G-Meta | 30.36 | 26.95 | 33.19 | 29.62 | 35.29 | 33.16 | 36.21 | 35.20 | 35.22 | 30.16 | 37.22 | 30.29 | 40.19 | 32.29 | 41.19 | 36.96 |
| Meta-GPS | 32.02 | 27.07 | 34.15 | 30.19 | 35.66 | 34.15 | 39.26 | 37.55 | 45.59 | 43.29 | 47.62 | 45.10 | 50.19 | 47.12 | 52.19 | 49.32 |
| X-FNC | 33.59 | 31.10 | 35.15 | 35.09 | 37.25 | 34.12 | 39.72 | 36.29 | 47.26 | 45.16 | 49.30 | 46.22 | 52.20 | 49.29 | 53.72 | 50.22 |
| COSMIC | 38.02 | 36.22 | 40.09 | 37.05 | 42.20 | 39.09 | 42.46 | 40.30 | 49.20 | 47.19 | 52.02 | 51.29 | 53.09 | 52.16 | 55.39 | 53.90 |
| TLP | 37.99 | 37.29 | 41.23 | 39.59 | 41.99 | 40.92 | 42.26 | 41.25 | 51.12 | 50.15 | 53.90 | 52.29 | 54.26 | 52.66 | 55.20 | 53.30 |
| TEG | 33.05 | 31.29 | 35.26 | 34.32 | 35.80 | 34.69 | 36.35 | 35.36 | 41.09 | 40.20 | 42.12 | 41.39 | 43.72 | 42.60 | 46.56 | 43.87 |
| SMILE | **42.64** | **41.27** | **45.14** | **43.69** | **45.88** | **44.10** | **46.72** | **45.65** | **56.36** | **55.25** | **58.84** | **57.53** | **59.08** | **57.96** | **59.38** | **58.25** |

| Dataset | Amazon-Electronics→DBLP | | | | | | | | DBLP→Amazon-Electronics | | | | | | | |
| | 5 tasks | | 10 tasks | | 15 tasks | | 20 tasks | | 5 tasks | | 10 tasks | | 15 tasks | | 20 tasks | |
| | Acc | F1 | Acc | F1 | Acc | F1 | Acc | F1 | Acc | F1 | Acc | F1 | Acc | F1 | Acc | F1 |
|---|---|---|---|---|---|---|---|---|---|---|---|---|---|---|---|---|
| Protonet | 31.86 | 22.56 | 32.58 | 23.73 | 35.90 | 32.76 | 39.88 | 35.71 | 28.84 | 18.62 | 30.54 | 20.08 | 33.10 | 22.37 | 35.46 | 25.20 |
| MAML | 29.17 | 19.13 | 30.10 | 22.15 | 32.97 | 25.98 | 35.25 | 29.11 | 26.59 | 17.99 | 28.36 | 19.29 | 30.02 | 20.15 | 32.16 | 22.16 |
| MetaMix | 40.16 | 35.68 | 43.25 | 41.69 | 45.19 | 43.12 | 49.12 | 43.59 | 37.70 | 35.22 | 40.20 | 39.09 | 42.25 | 40.16 | 44.19 | 42.20 |
| MLTI | 42.12 | 37.19 | 46.39 | 45.06 | 49.19 | 47.25 | 51.35 | 50.39 | 38.22 | 36.96 | 41.39 | 40.25 | 43.39 | 42.05 | 46.12 | 45.09 |
| Meta-Inter | 46.19 | 45.12 | 48.15 | 46.79 | 51.29 | 49.76 | 53.18 | 51.02 | 41.50 | 40.15 | 43.19 | 41.10 | 45.20 | 43.35 | 47.15 | 45.05 |
| Meta-GNN | 39.19 | 34.72 | 42.26 | 39.16 | 43.96 | 39.55 | 45.66 | 42.19 | 35.72 | 33.20 | 39.59 | 38.62 | 40.39 | 39.29 | 41.26 | 40.22 |
| GPN | 60.08 | 58.75 | 61.92 | 61.58 | 63.19 | 62.60 | 63.99 | 63.10 | 42.99 | 41.46 | 46.36 | 44.73 | 47.09 | 45.42 | 47.52 | 45.76 |
| G-Meta | 45.72 | 43.32 | 47.22 | 45.09 | 47.96 | 45.99 | 49.56 | 48.39 | 37.22 | 35.93 | 40.19 | 39.52 | 42.35 | 40.19 | 43.62 | 42.19 |
| Meta-GPS | 47.59 | 46.70 | 49.20 | 47.16 | 50.26 | 49.96 | 52.39 | 51.22 | 43.06 | 42.05 | 45.12 | 43.16 | 46.02 | 44.95 | 46.79 | 45.02 |
| X-FNC | 49.19 | 48.36 | 49.55 | 48.02 | 51.35 | 50.26 | 52.90 | 51.39 | 41.59 | 40.02 | 42.36 | 42.19 | 44.16 | 43.16 | 46.39 | 45.25 |
| COSMIC | 57.22 | 55.30 | 58.29 | 57.35 | 60.20 | 61.19 | 61.36 | 62.30 | 39.20 | 37.11 | 41.02 | 40.22 | 43.03 | 42.22 | 44.32 | 43.26 |
| TLP | 58.25 | 57.19 | 59.33 | 59.01 | 61.29 | 60.02 | 62.16 | 61.25 | 41.11 | 40.16 | 43.20 | 42.25 | 44.16 | 42.32 | 45.20 | 43.90 |
| TEG | 38.05 | 36.29 | 40.21 | 39.32 | 42.80 | 41.69 | 43.35 | 42.36 | 33.19 | 31.20 | 34.22 | 33.52 | 35.70 | 34.62 | 36.55 | 35.37 |
| SMILE | **62.44** | **61.66** | **64.54** | **64.16** | **65.04** | **64.43** | **65.78** | **65.42** | **46.24** | **44.54** | **48.82** | **47.26** | **49.26** | **47.70** | **49.52** | **47.88** |

**Table 4: Results of different model variants with respect to 5 tasks under the 5-way 5-shot setting.**

| Dataset | Clothing | | Electronics | | DBLP | | CoraFull | | Clothing→CoraFull | | CoraFull→Clothing | | Electronics→DBLP | | DBLP→Electronics | |
| | Acc | F1 | Acc | F1 | Acc | F1 | Acc | F1 | Acc | F1 | Acc | F1 | Acc | F1 | Acc | F1 |
|---|---|---|---|---|---|---|---|---|---|---|---|---|---|---|---|---|
| vanilla mixup | 78.82 | 78.45 | 60.98 | 60.70 | 74.06 | 73.03 | 61.90 | 61.07 | 38.90 | 38.26 | 52.93 | 52.29 | 57.92 | 57.30 | 40.52 | 39.26 |
| internal mixup | 78.96 | 78.72 | 63.59 | 62.21 | 75.34 | 74.45 | 63.64 | 63.00 | 39.12 | 38.59 | 53.22 | 53.19 | 59.02 | 57.96 | 41.36 | 40.56 |
| w/o within and across | 79.10 | 78.97 | 62.36 | 61.08 | 73.14 | 72.36 | 62.44 | 61.85 | 39.72 | 39.19 | 53.58 | 52.42 | 59.34 | 58.94 | 42.12 | 41.41 |
| w/o within | 80.72 | 80.93 | 65.96 | 64.99 | 75.44 | 74.59 | 64.06 | 63.40 | 40.28 | 39.63 | 54.52 | 53.50 | 60.86 | 59.16 | 42.28 | 41.79 |
| w/o across | 81.18 | 80.19 | 64.40 | 63.57 | 74.67 | 73.68 | 63.08 | 62.56 | 41.88 | 41.38 | 54.76 | 53.92 | 61.52 | 61.26 | 43.84 | 42.29 |
| w/o degree | 79.14 | 80.16 | 63.49 | 62.12 | 74.36 | 73.28 | 63.68 | 62.96 | 30.89 | 21.05 | 32.82 | 23.07 | 31.12 | 21.61 | 30.44 | 21.08 |
| ours | **82.80** | **82.49** | **67.30** | **66.30** | **75.88** | **75.05** | **66.34** | **65.70** | **42.64** | **41.27** | **56.36** | **55.25** | **62.44** | **61.66** | **46.24** | **44.54** |

data distribution, resulting in limited information gain. Moreover, we also provide the visualization study in **Appendix** G.

## 8 Conclusion

In this work, we propose a simple yet effective approach, called SMILE, for graph few-shot learning with fewer tasks. Specifically, we introduce a novel dual-level mixup strategy, including within-task and across-task mixup, for enriching the diversity of nodes within each task and the diversity of tasks. Also, we incorporate the degree-based prior information to learn expressive node embeddings. Theoretically, we prove that SMILE effectively enhances the model's generalization performance. Empirically, we conduct extensive experiments on multiple benchmarks and the results suggest that SMILE significantly outperforms other baselines, including both in-domain and cross-domain few-shot settings.

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

# Appendix

# A Description of Symbols

**Table S1: Descriptions of the symbols.**

| Symbols | Descriptions |
|---|---|
| $\mathcal{G}, \mathcal{V}, \mathcal{E}$ | Graph, node set, and edge set |
| $Z, A$ | Initialized node features and adjacency matrix |
| $\hat{D}, H, X$ | Degree matrix, hidden vectors, and refined vectors |
| $\kappa, \alpha, \beta$ | Interaction weights, node centralities, and adjusted scores |
| $N, K, M$ | $N$ way, $K$ shot, $M$ query |
| $\mathcal{D}_{org}$ | Original meta-training tasks |
| $\mathcal{D}_{aug}$ | Generated meta-training tasks |
| $\mathcal{D}_{all}$ | All original and generated meta-training tasks |
| $\mathcal{S}_t, \mathcal{Q}_t$ | Support and query set |
| $n_s, n_q$ | Number of samples in $\mathcal{S}_t$ and $\mathcal{Q}_t$ |
| $\mathcal{T}_{tes}$ | Meta-testing task |
| $\mathcal{S}_{tes}, \mathcal{Q}_{tes}$ | Support and query set of $\mathcal{T}_{tes}$ |
| $\eta, \zeta$ | Hyperparameters in Beta distribution |
| $\lambda$ | Random variable drawn from Beta distribution |
| $\mathcal{S}'_t, \mathcal{Q}'_t$ | Generated support and query set |
| $n_{s'}, n_{q'}$ | Number of samples in $\mathcal{S}'_t$ and $\mathcal{Q}'_t$ |
| $m', m$ | Number of samples in $\mathcal{S}_t \cup \mathcal{S}'_t$ and $\mathcal{Q}_t \cup \mathcal{Q}'_t$ |
| $\mathcal{T}_t^{aug}, \tilde{\mathcal{S}}, \tilde{\mathcal{Q}}$ | Interpolated task with its support and query set |
| $T_{org}$ | Number of tasks in $\mathcal{D}_{org}$ |
| $T_{aug}$ | Number of tasks in $\mathcal{D}_{aug}$ |
| $T$ | Number of tasks in $\mathcal{D}_{all}$ |

We summarize the used important symbols in Table S1.

# B Complexity Analysis

We analyze the time complexity of our proposed model to demonstrate its effectiveness. Our model mainly contains two parts, including node presentation learning and dual-level mixup. As linear interpolation is employed in the dual-level mixup, it does not introduce additional time complexity. Basically, most of the time-consuming operations arise from the node embedding process. Here, we choose SGC as the base graph encoder, which removes layer-wise non-linear operations and performs feature extraction in a parameter-free manner. The required time complexity of this step is $O(n^2 d)$, where $n$ and $d$ denote the number of nodes and the dimension of node features, respectively. Note that as feature extraction does not require any weights, it is essentially equivalent to a pre-processing step and can be precomputed in practice. Moreover, in the procedure of incorporating degree-based prior information to obtain the refined node representations, the required time complexity is $O(2nd + n)$. Thus, the overall time complexity of our approach is $O(n^2 d) + O(2nd + n)$, which is acceptable to us.

# C Theoretical Proof

## C.1 Proof of Eqs.9 and 10

To prove Eqs.9 and 10, we give the following Lemma C.1.

**Lemma C.1.** *Suppose the designed model with mixup distribution* $\lambda \sim Beta(\eta, \gamma)$. *Let* $\rho_\lambda \sim \frac{\eta}{\eta + \gamma} Beta(\eta + 1, \gamma) + \frac{\gamma}{\eta + \gamma}(\gamma + 1, \eta)$. *The approximation of the loss function* $\mathcal{L}(\mathcal{D}_{all}; \theta)$ *is given by,*

$$\mathcal{L}(\mathcal{D}_{all}; \theta) \approx \mathcal{L}(\mathcal{D}_{org}; \theta) + \mathcal{L}(\bar{\lambda}\mathcal{D}_{org}; \theta) + \mathcal{M}(\theta), \quad (16)$$

*where* $\bar{\lambda} = \mathbb{E}_{\rho_\lambda}[\lambda]$ *and* $\mathcal{M}(\theta)$ *is a quadratic regularization term with respect to* $\theta$, *defined as*

$$\mathcal{M}(\theta) = \mathbb{E}_{\mathcal{T}_t \sim p(\mathcal{T})} \mathbb{E}_{(X_t, Y_t) \sim q(\mathcal{T}_t)} \frac{\phi(P_t)(\phi(P_t) - 0.5)}{2(1 + \exp(P_t))} (\theta^\top \Sigma_X \theta) \quad (17)$$

*in which* $P_t = \langle X_t^q - (C_1 + C_2)/2, \theta \rangle$, $\phi(P_t) = \exp(P_t)/(1 + \exp(P_t))$, *and* $\Sigma_X = \mathbb{E}[XX^\top] = \frac{1}{m} \sum_{i=1}^m X_i X_i^\top$.

PROOF. As stated in Section 5, the adopted simplified loss function for binary classification can be formulated as:

$$\mathcal{L}(\mathcal{D}_{all}; \theta) = \mathcal{L}(\mathcal{D}_{org}; \theta) + \mathcal{L}(\mathcal{D}_{aug}; \theta)$$

$$= \mathcal{L}(\{\mathcal{T}_t\}_{t=1}^{T_{org}}; \theta) + \mathcal{L}(\{\mathcal{T}_t^{aug}\}_{t=1}^{T_{aug}}; \theta)$$

$$= \mathbb{E}_{\mathcal{T}_t \sim p(\mathcal{T})} \mathbb{E}_{(X_t, Y_t) \sim q(\mathcal{T}_t)} (1 + \exp(\langle X_t^q - (C_1 + C_2)/2, \theta \rangle))^{-1} +$$

$$\mathbb{E}_{\mathcal{T}_t^{aug} \sim p(\mathcal{T}^{aug})} \mathbb{E}_{(\tilde{X}_t, \tilde{Y}_t) \sim q(\mathcal{T}_t^{aug})} (1 + \exp(\langle \tilde{X}_t^q - (\tilde{C}_1 + \tilde{C}_2)/2, \theta \rangle))^{-1}, \quad (18)$$

where

$$C_k = \frac{1}{|\mathcal{S}_{t,k}|} \sum_{(X_{t,i}^s, Y_{t,i}^s) \in \mathcal{S}_t} \mathbb{I}_{Y_{t,i} = k} X_{t,i}^s,$$

$$\tilde{C}_k = \frac{1}{|\tilde{\mathcal{S}}_{t,k}|} \sum_{(\tilde{X}_{t,i}^s, \tilde{Y}_{t,i}^s) \in \tilde{\mathcal{S}}_t} \mathbb{I}_{\tilde{Y}_{t,i} = k} \tilde{X}_{t,i}^s. \quad (19)$$

Since the preprocessed centralized dataset satisfies the condition $\mathbb{E}_{\mathcal{T}_t \sim p(\mathcal{T})} \mathbb{E}_{(X_t, Y_t) \sim q(\mathcal{T}_t)} X_t = \frac{1}{T_{org}} \sum_{t=1}^{T_{org}} \frac{1}{2n_q} \sum_{k=1}^2 \sum_{i=1}^{n_q} \theta^\top X_{t,i;k} = 0$, which means that the overall sample mean equals 0, we can obtain $\frac{1}{\lambda} \mathbb{E}[\tilde{X}_{t,i;k} | X_{t,i;k}] = X_{t,i;k}$. Moreover, as we simultaneously include linear weights and biases, the predictions are invariant to scaling and shifting of $\tilde{X}_t$, so it suffices to consider $\{\tilde{X}_t, \tilde{Y}_t\}_{t=1}^{T_{aug}}$ with $\tilde{X} = \frac{1}{\lambda}(\lambda X_{t;k} + (1 - \lambda) X_{m;k})$. Then, we apply the second-order Taylor expansion on $\mathcal{L}(\mathcal{D}_{aug}; \theta) = \mathcal{L}(\{\mathcal{T}_t^{aug}\}_{t=1}^{T_{aug}}; \theta) = \mathbb{E}_{\mathcal{T}_t^{aug} \sim p(\mathcal{T}^{aug})} \mathbb{E}_{(\tilde{X}_t, \tilde{Y}_t) \sim q(\mathcal{T}_t^{aug})} (1 + \exp(\langle \tilde{X}_t^q - (\tilde{C}_1 + \tilde{C}_2)/2, \theta \rangle))^{-1}$ with respect to $\tilde{X}_t^q$ around $\frac{1}{\lambda} \mathbb{E}[\tilde{X}_{t,i;k}^q | X_{t,i;k}^q] = \theta^\top X_{t,i;k}^q$ as follows.

$$\mathcal{L}(\mathcal{D}_{aug}; \theta) \approx \mathcal{L}(\bar{\lambda}\mathcal{D}_{org}; \theta) + \frac{\partial \mathcal{L}(\mathcal{D}_{org}; \theta)}{\partial \theta^\top X_t^q} (\theta^\top \tilde{X}_t^q - \theta^\top X_t^q)$$

$$+ (\theta^\top \tilde{X}_t^q - \theta^\top X_t^q)^\top \frac{\partial^2 \mathcal{L}(\mathcal{D}_{org}; \theta)}{\partial (\theta^\top X_t^q)^2} (\theta^\top \tilde{X}_t^q - \theta^\top X_t^q) \quad (20)$$

For ease of presentation, let $H_t = \theta^\top X_t^q$ and $P_t = \langle X_t^q - (C_1 + C_2)/2, \theta \rangle$. Then, we have

$$\frac{\partial \mathcal{L}(\mathcal{D}_{org}; \theta)}{\partial H_t} = \frac{\partial \mathcal{L}(\mathcal{D}_{org}; \theta)}{\partial P_t} \times \frac{\partial P_t}{\partial H_t}$$

$$= \mathbb{E}_{\mathcal{T}_t \sim p(\mathcal{T})} \mathbb{E}_{(X_t, Y_t) \sim q(\mathcal{T}_t)} \frac{\exp(P_t)}{2(1 + \exp(P_t))^2} \quad (21)$$

By defining the function $\phi(P_t) = \frac{\exp(P_t)}{1+\exp(P_t)}$, we have

$$\frac{\partial^2 \mathcal{L}(\mathcal{D}_{org};\theta)}{\partial H_t^2} = \mathbb{E}_{\mathcal{T}_t \sim p(\mathcal{T})}\mathbb{E}_{(X_t,Y_t)\sim q(\mathcal{T}_t)}$$

$$\left[ -\frac{\exp(P_t)}{(1+\exp(P_t))^3}\frac{\exp(P_t)}{\partial H_t} + \frac{1}{2(1+\exp(P_t))^2}\frac{\partial \exp(P_t)}{\partial H_t} \right]$$

$$= \mathbb{E}_{\mathcal{T}_t \sim p(\mathcal{T})}\mathbb{E}_{(X_t,Y_t)\sim q(\mathcal{T}_t)}\left[ \frac{\exp(P_t)^2}{2(1+\exp(P_t))^3} - \frac{\exp(P_t)}{4(1+\exp(P_t))^2} \right]$$

$$= \mathbb{E}_{\mathcal{T}_t \sim p(\mathcal{T})}\mathbb{E}_{(X_t,Y_t)\sim q(\mathcal{T}_t)} \frac{\exp(P_t)(\phi(P_t)-0.5)}{2(1+\exp(P_t))^2}$$

$$= \mathbb{E}_{\mathcal{T}_t \sim p(\mathcal{T})}\mathbb{E}_{(X_t,Y_t)\sim q(\mathcal{T}_t)} \frac{\phi(P_t)(\phi(P_t)-0.5)}{2(1+\exp(P_t))} \tag{22}$$

Plugging Eqs.21 and 22 into the Eq.20, we can obtain

$$\mathcal{L}(\mathcal{D}_{aug};\theta) \approx \mathcal{L}(\bar{\lambda}\mathcal{D}_{org};\theta)+$$

$$\mathbb{E}_{\mathcal{T}_t \sim p(\mathcal{T})}\mathbb{E}_{(X_t,Y_t)\sim q(\mathcal{T}_t)} \frac{\exp(P_t)}{2(1+\exp(P_t))^2}(\theta^\top \tilde{X}_t^q - \theta^\top X_t^q)+$$

$$\left( (\theta^\top \tilde{X}_t^q - \theta^\top X_t^q)^\top \mathbb{E}_{\mathcal{T}_t \sim p(\mathcal{T})}\mathbb{E}_{(X_t,Y_t)\sim q(\mathcal{T}_t)} \right. \tag{23}$$

$$\left. \frac{\phi(P_t)(\phi(P_t)-0.5)}{2(1+\exp(P_t))}(\theta^\top \tilde{X}_t^q - \theta^\top X_t^q) \right)$$

Since $\mathbb{E}[\tilde{X}_t - X_t] = 0$ and $\mathrm{Var}(\tilde{X}_t) = \mathbb{E}[XX^\top] = \Sigma_X$, the first-order term vanishes and we then simplify the above equation as

$$\mathcal{L}(\mathcal{D}_{aug};\theta) \approx \mathcal{L}(\bar{\lambda}\mathcal{D}_{org};\theta)$$

$$+ \mathbb{E}_{\mathcal{T}_t \sim p(\mathcal{T})}\mathbb{E}_{(X_t,Y_t)\sim q(\mathcal{T}_t)} \frac{\phi(P_t)(\phi(P_t)-0.5)}{2(1+\exp(P_t))}(\theta^\top \Sigma_X \theta) \tag{24}$$

Combining Eqs.18 and 24, we can acquire

$$\mathcal{L}(\mathcal{D}_{all};\theta) = \mathcal{L}(\mathcal{D}_{org};\theta) + \mathcal{L}(\mathcal{D}_{aug};\theta)$$

$$\approx \mathcal{L}(\mathcal{D}_{org};\theta) + \mathcal{L}(\bar{\lambda}\mathcal{D}_{org};\theta)$$

$$+ \mathbb{E}_{\mathcal{T}_t \sim p(\mathcal{T})}\mathbb{E}_{(X_t,Y_t)\sim q(\mathcal{T}_t)} \frac{\phi(P_t)(\phi(P_t)-0.5)}{2(1+\exp(P_t))}(\theta^\top \Sigma_X \theta) \tag{25}$$

$$= \mathcal{L}(\mathcal{D}_{org};\theta) + \mathcal{L}(\bar{\lambda}\mathcal{D}_{org};\theta) + \mathcal{M}(\theta)$$

Thus, we complete the proof and derive the desired results. □

## C.2 Proof of Theorem 5.1

Before formally proving Theorem 5.1, we first provide some relevant definitions. The *Rademacher complexity* [30] reflects the richness of a function class by measuring the extent to which the hypothesis set fits random noise. It is a commonly used and flexible measure of complexity for hypothesis classes, which is defined formally as follows.

**Definition C.2. Empirical Rademacher Complexity**. *The empirical Rademacher complexity of a function class $\mathcal{F}$ with respect to a sample set $\{z_i\}_{i=1}^m$ of size m drawn from a specific data distribution is defined as:*

$$\hat{\mathcal{R}}(\mathcal{F}|z_1,\ldots,z_m) = \mathbb{E}_\sigma\left[ \sup_{f\in\mathcal{F}}\frac{1}{m}\sum_{i=1}^m \sigma_i f(z_i) \right], \tag{26}$$

*where $\sigma = [\sigma_1,\ldots,\sigma_m]^\top$ are Rademacher variables, in which $\sigma_i$ follows a uniform distribution and takes values in $\{-1,+1\}$.*

Next, we present the definition of *Rademacher complexity*, which eliminates the dependence on specific sample sets and provides a more uniform measure of the complexity of a function class.

**Definition C.3. Rademacher Complexity**. *The Rademacher complexity of a function class $\mathcal{F}$ is defined by the expectation of the empirical Rademacher complexity over all samples of size m drawn according to data distribution $\mathbb{P}$:*

$$\mathcal{R}(\mathcal{F}) = \mathbb{E}_{\{z_1,\ldots,z_m\}\sim\mathbb{P}}\hat{\mathcal{R}}(\mathcal{F}|z_1,\ldots,z_m). \tag{27}$$

The *integral probability metric* [32] is a type of distance function between probability distributions that measures how a class of functions distinguishes between two probability distributions. Its formal definition is as follows.

**Definition C.4. Integral Probability Metric**. *The integral probability metric between two probability distributions $\mathbb{P}$ and $\mathbb{Q}$ on the data space $\mathcal{Z}$ with respect to the class of real-valued bounded measurable functions $\mathcal{F}$ is given by:*

$$\mathcal{D}_\mathcal{F}(\mathbb{P},\mathbb{Q}) = \sup_{f\in\mathcal{F}}\left| \mathbb{E}_\mathbb{P}f(\mathcal{Z}) - \mathbb{E}_\mathbb{Q}f(\mathcal{Z}) \right|. \tag{28}$$

Next, we present the standard uniform deviation bound in Lemma C.5 [30] using Rademacher complexity.

**Lemma C.5.** *Let $\mathcal{F}$ be a collection of functions mapping from $\mathcal{Z}$ to $[a, a+1]$. For $\epsilon > 0$, with probability at least $(1-\epsilon)$ over an i.i.d. sample set $\{z_i\}_{i=1}^m$ drawn from a distribution $\mathbb{P}$ over $\mathcal{Z}$, each $f$ in $\mathcal{F}$ satisfies:*

$$\mathbb{E}_\mathbb{P}f(z) \le \frac{1}{m}\sum_{i=1}^m f(z_i) + 2\mathcal{R}(\mathcal{F}) + \sqrt{\frac{\log(1/\epsilon)}{2m}}, \text{ or}$$

$$\mathbb{E}_\mathbb{P}f(z) \le \frac{1}{m}\sum_{i=1}^m f(z_i) + 2\hat{\mathcal{R}}(\mathcal{F}|z_1,\cdots,z_m) + 3\sqrt{\frac{\log(2/\epsilon)}{2m}}, \tag{29}$$

*where $\mathcal{R}(\mathcal{F})$ and $\hat{\mathcal{R}}(\mathcal{F}|z_1,\cdots,z_m)$ are the Rademacher complexity and the empirical Rademacher complexity, respectively.*

We can leverage the integral probability metric to transform Eq.29 into the following form:

$$\mathcal{D}_\mathcal{F}(\hat{\mathbb{P}},\mathbb{P}) = \sup_{f\in\mathcal{F}}\left| \mathbb{E}_{\hat{\mathbb{P}}}f(z) - \mathbb{E}_\mathbb{P}f(z) \right| \le 2\mathcal{R}(\mathcal{F}) + \sqrt{\frac{\log(1/\epsilon)}{2m}}, or$$

$$\mathcal{D}_\mathcal{F}(\hat{\mathbb{P}},\mathbb{P}) = \sup_{f\in\mathcal{F}}\left| \mathbb{E}_{\hat{\mathbb{P}}}f(z) - \mathbb{E}_\mathbb{P}f(z) \right| \le 2\hat{\mathcal{R}}(\mathcal{F}|z_1,\cdots,z_m) + 3\sqrt{\frac{\log(2/\epsilon)}{2m}}, \tag{30}$$

where $\hat{\mathbb{P}}$ denotes the empirical distribution of samples.

Additionally, we introduce Lemma C.6 that bounds the Rademacher complexity, which is utilized in the proof of Theorem 5.1.

**Lemma C.6.** *Let $\sum_X = \mathbb{E}[XX^\top]$ and $\mathcal{F}_v = \{X \to \theta^\top X : \theta^\top \sum_X \theta \le v\}$. Then, the Rademacher complexity of $\mathcal{F}_v$ satisfies*

$$\mathcal{R}(\mathcal{F}_v) \le \sqrt{\frac{v \cdot rank(\sum_X)}{m}}. \tag{31}$$

PROOF. According to the definition of empirical Rademacher complexity, given a set of *i.i.d* Rademacher random variables, we

have:

$$\hat{R}(\mathcal{F}|X_1,\ldots,X_m) = \mathbb{E}_\sigma\left[\sup_{f\in\mathcal{F}}\frac{1}{m}\sum_{i=1}^m \sigma_i f(X_i)\right]$$

$$= \mathbb{E}_\sigma\left[\sup_{\theta^\top\Sigma_X\theta\le v}\frac{1}{m}\sum_{i=1}^m \sigma_i f(X_i)\right]$$

$$= \mathbb{E}_\sigma\left[\sup_{\theta^\top\Sigma_X\theta\le v}\frac{1}{m}\sum_{i=1}^m \sigma_i\theta^\top X_i\right]$$

$$= \mathbb{E}_\sigma\left[\sup_{||\Sigma_X^{1/2}\theta||^2\le v}\frac{1}{m}\sum_{i=1}^m \sigma_i(\Sigma_X^{1/2}\theta)^\top(\Sigma_X^{\dagger/2}X_i)\right]$$

$$\le \frac{\sqrt{v}}{m}\mathbb{E}_\sigma\left\|\sum_{i=1}^m \sigma_i\Sigma_X^{\dagger/2}X_i\right\|$$

$$\le \frac{\sqrt{v}}{m}\sqrt{\mathbb{E}_\sigma\left\|\sum_{i=1}^m \sigma_i\Sigma_X^{\dagger/2}X_i\right\|^2}$$

$$\le \frac{\sqrt{v}}{m}\sqrt{\sum_{i=1}^m (\Sigma_X^{\dagger/2}X_i)^\top(\Sigma_X^{\dagger/2}X_i)}$$

$$= \frac{\sqrt{v}}{m}\sqrt{\sum_{i=1}^m X_i^\top X_i},$$

where $\Sigma_X^\dagger$ is the generalized inverse of $\Sigma_X$.

$$\mathcal{R}(\mathcal{F}) = \mathbb{E}_{\{X_1,\ldots,X_m\}\sim\mathbb{P}}\hat{R}(\mathcal{F}|X_1,\ldots,X_m)$$

$$\le \frac{\sqrt{v}}{m}\sqrt{\sum_{i=1}^m \mathbb{E}_{X_i}(X_i^\top X_i)}$$

$$\le \frac{\sqrt{v}\sqrt{\text{rank}(\Sigma_{X_i})}}{\sqrt{m}}.$$

$\square$

Now, we are ready to prove the Theorem 5.1.

Proof.

$$|\hat{R} - R|$$

$$= \left|\mathbb{E}_{\mathcal{T}_i\sim\hat{p}(\mathcal{T})}\mathbb{E}_{(X_j,Y_j)\sim\hat{q}(\mathcal{T}_i)}\mathcal{L}(f_\theta(X_j),Y_j) - \mathbb{E}_{\mathcal{T}_i\sim p(\mathcal{T})}\mathbb{E}_{(X_j,Y_j)\sim q(\mathcal{T}_i)}\mathcal{L}(f_\theta(X_j),Y_j)\right|$$

$$= \left|\mathbb{E}_{\mathcal{T}_i\sim\hat{p}(\mathcal{T})}\mathbb{E}_{(X_j,Y_j)\sim\hat{q}(\mathcal{T}_i)}\mathcal{L}(f_\theta(X_j),Y_j) - \right.$$
$$\mathbb{E}_{\mathcal{T}_i\sim\hat{p}(\mathcal{T})}\mathbb{E}_{(X_j,Y_j)\sim q(\mathcal{T}_i)}\mathcal{L}(f_\theta(X_j),Y_j) +$$
$$\mathbb{E}_{\mathcal{T}_i\sim\hat{p}(\mathcal{T})}\mathbb{E}_{(X_j,Y_j)\sim q(\mathcal{T}_i)}\mathcal{L}(f_\theta(X_j),Y_j) -$$
$$\left.\mathbb{E}_{\mathcal{T}_i\sim p(\mathcal{T})}\mathbb{E}_{(X_j,Y_j)\sim q(\mathcal{T}_i)}\mathcal{L}(f_\theta(X_j),Y_j)\right| \quad (32)$$

For the *first two terms* of Eq.32, we can rewrite them as:

$$\mathbb{E}_{\mathcal{T}_i\sim\hat{p}(\mathcal{T})}\mathbb{E}_{(X_j,Y_j)\sim\hat{q}(\mathcal{T}_i)}\mathcal{L}(f_\theta(X_j),Y_j) -$$
$$\mathbb{E}_{\mathcal{T}_i\sim\hat{p}(\mathcal{T})}\mathbb{E}_{(X_j,Y_j)\sim q(\mathcal{T}_i)}\mathcal{L}(f_\theta(X_j),Y_j)$$

$$\le \mathbb{E}_{\mathcal{T}_i\sim\hat{p}(\mathcal{T})}\left[\mathbb{E}_{(X_j,Y_j)\sim\hat{q}(\mathcal{T}_i)}\mathcal{L}(f_\theta(X_j),Y_j) - \right.$$
$$\left.\mathbb{E}_{(X_j,Y_j)\sim q(\mathcal{T}_i)}\mathcal{L}(f_\theta(X_j),Y_j)\right]$$

$$\overset{(i)}{\le} \mathbb{E}_{\mathcal{T}_i\sim\hat{p}(\mathcal{T})}[\mathcal{D}_\mathcal{F}(\hat{q},q)]$$

$$\overset{(ii)}{=} \mathbb{E}_{\mathcal{T}_i\sim\hat{p}(\mathcal{T})}\left[\sup_{f\in\mathcal{F}}\left|\mathbb{E}_{\hat{q}}\mathcal{L}(f_\theta(X_j),Y_j) - \mathbb{E}_q\mathcal{L}(f_\theta(X_j),Y_j)\right|\right] \quad (33)$$

$$\overset{(iii)}{\le} \mathbb{E}_{\mathcal{T}_i\sim\hat{p}(\mathcal{T})}\left[2\hat{R}(\mathcal{F}|X_1,\cdots,X_m) + 3\sqrt{\frac{\log(2/\epsilon)}{2m}}\right]$$

$$\overset{(iv)}{=} 2\mathcal{R}(\mathcal{F}) + 3\sqrt{\frac{\log(2/\epsilon)}{2m}}$$

$$\overset{(v)}{\le} 2\sqrt{\frac{v\cdot\text{rank}(\Sigma_X)}{m}} + 3\sqrt{\frac{\log(2/\epsilon)}{2m}},$$

where Inequality $(i)$ holds due to the definition of the integral probability metric, Equation $(ii)$ is the expansion of the integral probability metric, Inequality $(iii)$ holds due to Lemma C.5, Equation $(iv)$ is the definition of Rademacher complexity, and Inequality $(v)$ holds due to Lemma C.6.

For the *last two terms* of Eq.32, we first define a function class $\mathcal{H}$, which satisfies:

$$\mathcal{H} = \{h(\mathcal{T}) : h(\mathcal{T}) = \mathbb{E}_{(X_j,Y_j)\sim q(\mathcal{T})}(\mathcal{L}(f_\theta(X_j),Y_j),$$
$$\text{such that } f_\theta\in\mathcal{F}_v \text{ and } h(\cdot) \text{ maps } \mathcal{T} \text{ to } \mathbb{R}\}.$$

Then, by utilizing the integral probability metric and Lemma C.5, we have

$$\mathbb{E}_{\mathcal{T}_i\sim\hat{p}(\mathcal{T})}\mathbb{E}_{(X_j,Y_j)\sim q(\mathcal{T}_i)}\mathcal{L}(f_\theta(X_j),Y_j) -$$
$$\mathbb{E}_{\mathcal{T}_i\sim p(\mathcal{T})}\mathbb{E}_{(X_j,Y_j)\sim q(\mathcal{T}_i)}\mathcal{L}(f_\theta(X_j),Y_j)$$

$$= \mathbb{E}_{\mathcal{T}_i\sim\hat{p}(\mathcal{T})}h(\mathcal{T}_i) - \mathbb{E}_{\mathcal{T}_i\sim p(\mathcal{T})}h(\mathcal{T}_i)$$

$$= \mathcal{D}_\mathcal{H}(\hat{p},p) \quad (34)$$

$$= \sup_{h\in\mathcal{H}}\left|\mathbb{E}_{\hat{p}}h(\mathcal{T}_i) - \mathbb{E}_p h(\mathcal{T}_i)\right|$$

$$\le 2\hat{R}(\mathcal{H}|\mathcal{T}_1,\cdots,\mathcal{T}_T) + 3\sqrt{\frac{\log(2/\epsilon)}{2T}}.$$

Next, we need to obtain the empirical Rademacher complexity for the defined function class over distributions. According to the

definition, we have

$$\hat{\mathcal{R}}(\mathcal{H}|\mathcal{T}_1,\cdots,\mathcal{T}_T) = \mathbb{E}_{\sigma}\left[\sup_{h\in\mathcal{H}}\frac{1}{T}\sum_{i=1}^{T}\sigma_i h(\mathcal{T}_i)\right]$$

$$= \mathbb{E}_{\sigma}\left[\sup_{h\in\mathcal{H}}\frac{1}{T}\sum_{i=1}^{T}\sigma_i \mathbb{E}_{(X_j,Y_j)\sim q(\mathcal{T}_i)}\left(\frac{1}{1+\exp(\theta^{\top}X_j)}-\theta^{\top}X_jY_j\right)\right]$$

$$\leq \mathbb{E}_{\sigma}\left[\sup_{h\in\mathcal{H}}\frac{1}{T}\sum_{i=1}^{T}\sigma_i \mathbb{E}_{(X_j,Y_j)\sim q(\mathcal{T}_i)}\left(\theta^{\top}X_t\right)\right]+$$

$$\mathbb{E}_{\sigma}\left[\sup_{h\in\mathcal{H}}\frac{1}{T}\sum_{i=1}^{T}\sigma_i \mathbb{E}_{(X_j,Y_j)\sim q(\mathcal{T}_i)}\left(\theta^{\top}X_jY_j\right)\right]$$

$$\leq \underbrace{\mathbb{E}_{\sigma}\left[\sup_{h\in\mathcal{H}}\frac{1}{T}\sum_{i=1}^{T}\sigma_i (\Sigma_X^{1/2}\theta)^{\top}\Sigma_X^{\dagger/2}\mu_X\right]}_{(i)}+$$

$$\underbrace{\mathbb{E}_{\sigma}\left[\sup_{h\in\mathcal{H}}\frac{1}{T}(\Sigma_X^{1/2}\theta)^{\top}\sum_{i=1}^{T}\sigma_i \mathbb{E}_{(X_j,Y_j)\sim q(\mathcal{T}_i)}\left(\Sigma_X^{\dagger/2}X_jY_j\right)\right]}_{(ii)},$$

$$\tag{35}$$

where $\mu_X = \mathbb{E}_{(X_j,Y_j)\sim q(\mathcal{T}_i)}X_j$.

For the *first term (i)* in Eq.35, we can bound it as follows:

$$\mathbb{E}_{\sigma}\left[\sup_{h\in\mathcal{H}}\frac{1}{T}\sum_{i=1}^{T}\sigma_i (\Sigma_X^{1/2}\theta)^{\top}\Sigma_X^{\dagger/2}\mu_X\right]$$

$$\leq \left\|(\Sigma_X^{1/2}\theta)^{\top}\right\|\cdot\mathbb{E}_{\sigma}\left[\sup_{h\in\mathcal{H}}\frac{1}{T}\sum_{i=1}^{T}\sigma_i \Sigma_X^{\dagger/2}\mu_X\right]$$

$$\leq \sqrt{\nu}\mathbb{E}_{\sigma}\left\|\frac{1}{T}\sum_{i=1}^{T}\sigma_i \Sigma_X^{\dagger/2}\mu_X\right\| \leq \frac{\sqrt{\nu}}{T}\sqrt{\mathbb{E}_{\sigma}\left\|\sum_{i=1}^{T}\sigma_i \Sigma_X^{\dagger/2}\mu_X\right\|^2}$$

$$\tag{36}$$

$$\leq \frac{\sqrt{\nu}}{\sqrt{T}}\left\|\Sigma_X^{\dagger/2}\mu_X\right\|.$$

For the *second term (ii)* in Eq.35, we have the following bound as follows:

$$\mathbb{E}_{\sigma}\left[\sup_{h\in\mathcal{H}}\frac{1}{T}(\Sigma_X^{1/2}\theta)^{\top}\sum_{i=1}^{T}\sigma_i \mathbb{E}_{(X_j,Y_j)\sim q(\mathcal{T}_i)}\left(\Sigma_X^{\dagger/2}X_jY_j\right)\right]$$

$$\leq \left\|(\Sigma_X^{1/2}\theta)^{\top}\right\|\cdot\mathbb{E}_{\sigma}\left[\sup_{h\in\mathcal{H}}\frac{1}{T}\sum_{i=1}^{T}\sigma_i \mathbb{E}_{(X_j,Y_j)\sim q(\mathcal{T}_i)}\left(\Sigma_X^{\dagger/2}X_jY_j\right)\right]$$

$$\leq \sqrt{\nu}\mathbb{E}_{\sigma}\left\|\frac{1}{T}\sum_{i=1}^{T}\sigma_i \mathbb{E}_{(X_j,Y_j)\sim q(\mathcal{T}_i)}\left(\Sigma_X^{\dagger/2}X_jY_j\right)\right\|$$

$$\tag{37}$$

$$\leq \frac{\sqrt{\nu}}{T}\sqrt{\mathbb{E}_{\sigma}\left\|\sum_{i=1}^{T}\sigma_i \mathbb{E}_{(X_j,Y_j)\sim q(\mathcal{T}_i)}(\Sigma_X^{\dagger/2}X_jY_j)\right\|^2}$$

$$\leq \frac{\sqrt{\nu}}{\sqrt{T}}\sqrt{\sum_{i=1}^{T}\mathbb{E}_{(X_j,Y_j)\sim q(\mathcal{T}_i)}\left[(\Sigma_X^{\dagger/2}X_jY_j)^{\top}(\Sigma_X^{\dagger/2}X_jY_j)\right]}$$

$$\leq \frac{\sqrt{\nu}\cdot\text{rank}(\Sigma_X)}{\sqrt{T}}.$$

By combining the derivations from Eqs.36 and 37, we have the following bound in Eq.35:

$$\hat{\mathcal{R}}(\mathcal{H}|\mathcal{T}_1,\cdots,\mathcal{T}_T) \leq \sqrt{\frac{\nu}{T}}\left(\left\|\Sigma_X^{\dagger/2}\mu_X\right\|+\text{rank}(\Sigma_X)\right). \tag{38}$$

Thus, the last two terms of Eq.32 can be bounded as follows:

$$\mathbb{E}_{\mathcal{T}_i\sim\hat{p}(\mathcal{T})}\mathbb{E}_{(X_j,Y_j)\sim q(\mathcal{T}_i)}\mathcal{L}(f_\theta(X_j),Y_j)-$$
$$\mathbb{E}_{\mathcal{T}_i\sim p(\mathcal{T})}\mathbb{E}_{(X_j,Y_j)\sim q(\mathcal{T}_i)}\mathcal{L}(f_\theta(X_j),Y_j)$$

$$\leq 2\hat{\mathcal{R}}(\mathcal{H}|\mathcal{T}_1,\cdots,\mathcal{T}_T)+3\sqrt{\frac{\log(2/\epsilon)}{2T}} \tag{39}$$

$$\leq 2\sqrt{\frac{\nu}{T}}\left(\left\|\Sigma_X^{\dagger/2}\mu_X\right\|+\text{rank}(\Sigma_X)\right)+3\sqrt{\frac{\log(2/\epsilon)}{2T}}.$$

By combining the results from Eqs.33 and 39, we can obtain the following result.

$$|\hat{R}-R| \leq 2\left(\sqrt{\frac{\nu\cdot\text{rank}(\Sigma_X)}{m}}+\sqrt{\frac{\nu}{T}}\left(\left\|\Sigma_X^{\dagger/2}\mu_X\right\|+\text{rank}(\Sigma_X)\right)\right)+$$

$$3\left(\sqrt{\frac{\log(2/\epsilon)}{2m}}+\sqrt{\frac{\log(2/\epsilon)}{2T}}\right).$$

$$\tag{40}$$

When $\mu_X$ is set to 0, we can obtain the desired outcome as shown in Theorem 5.1, which is listed below.

$$|\hat{R}-R| \leq 2\left(\sqrt{\frac{\nu\cdot\text{rank}(\Sigma_X)}{m}}+\sqrt{\frac{\nu}{T}}\left(\text{rank}(\Sigma_X)\right)\right)+$$

$$3\left(\sqrt{\frac{\log(2/\epsilon)}{2m}}+\sqrt{\frac{\log(2/\epsilon)}{2T}}\right).$$

$$\tag{41}$$

$$\square$$

### C.3 Proof of Corollary 5.2

PROOF. According to Theorem 5.1, the upper bound of the model using our proposed strategy can be represented as:

$$U(|\hat{R}-R|) = 2\left(\sqrt{\frac{\nu\cdot\text{rank}(\Sigma_X)}{m}}+\sqrt{\frac{\nu}{T}}\left(\text{rank}(\Sigma_X)\right)\right)+$$

$$3\left(\sqrt{\frac{\log(2/\epsilon)}{2m}}+\sqrt{\frac{\log(2/\epsilon)}{2T}}\right).$$

$$\tag{42}$$

The upper bound of the model without any strategy can be represented as:

$$U(|\hat{R}_{\text{ori}}-R_{\text{ori}}|) = 2\left(\sqrt{\frac{\nu\cdot\text{rank}(\Sigma_X)}{m_{\text{ori}}}}+\sqrt{\frac{\nu}{T_{\text{ori}}}}\left(\text{rank}(\Sigma_X)\right)\right)+$$

$$3\left(\sqrt{\frac{\log(2/\epsilon)}{2m_{\text{ori}}}}+\sqrt{\frac{\log(2/\epsilon)}{2T_{\text{ori}}}}\right).$$

$$\tag{43}$$

Thus, we can proceed with the proof as follows:

$$\mathsf{U}(|\hat{\mathsf{R}} - \mathsf{R}|) - \mathsf{U}(|\hat{\mathsf{R}}_{\mathrm{ori}} - \mathsf{R}_{\mathrm{ori}}|)$$

$$= 2 \left[ \sqrt{\nu \cdot \mathrm{rank}(\Sigma_{\mathrm{X}})} \left( \underbrace{\sqrt{\frac{1}{m}} - \sqrt{\frac{1}{m_{\mathrm{ori}}}}}_{(i)} \right) + \right.$$

$$\left. \left( \sqrt{\nu \, (\mathrm{rank}(\Sigma_{\mathrm{X}}))} \left( \underbrace{\sqrt{\frac{1}{\mathsf{T}}} - \sqrt{\frac{1}{\mathsf{T}_{\mathrm{ori}}}} \mathsf{T}_{\mathrm{ori}}}_{(ii)} \right) \right) \right]$$

$$+ 3 \left[ \sqrt{\log(2/\epsilon)} \left( \underbrace{\sqrt{\frac{1}{2m}} - \sqrt{\frac{1}{2m_{\mathrm{ori}}}}}_{(iii)} \right) + \sqrt{\log(2/\epsilon)} \left( \underbrace{\sqrt{\frac{1}{2\mathsf{T}}} - \sqrt{\frac{1}{2\mathsf{T}_{\mathrm{ori}}}}}_{(iv)} \right) \right].$$

$$(44)$$

Due to the introduction of within-task interpolation, we have $m > m_{\mathrm{ori}}$, thus inequalities $(i)$ and $(iii)$ are both less than 0. Furthermore, due to the introduction of inter-task interpolation, we have $T > T_{\mathrm{ori}}$, thus inequalities $(ii)$ and $(iv)$ are also both less than 0. Therefore, the inequality $\mathsf{U}(|\hat{\mathsf{R}} - \mathsf{R}|) \le \mathsf{U}(|\hat{\mathsf{R}}_{\mathrm{ori}} - \mathsf{R}_{\mathrm{ori}}|)$ is held. The Corollary 5.2 is also satisfied. $\qquad\square$

## C.4 Proof of Theorem 5.3

According to Theorem 5.3, we have

$$\sup_{f \in \mathcal{F}} |\mathbb{E}_{\hat{\mathbb{P}}} - \mathbb{E}_{\mathbb{Q}}| \le \left( 2\sqrt{\nu \cdot \mathrm{rank}(\Sigma_{\mathrm{X}})} + \sqrt{\frac{\log(1/\epsilon)}{2}} \right) \left( \sqrt{\frac{1}{m}} + \sqrt{\frac{1}{n_q}} \right). \quad (45)$$

The procedure of Theorem 5.3 is summarized as follows.

PROOF.

$$\sup_{f \in \mathcal{F}} |\mathbb{E}_{\hat{\mathbb{P}}} - \mathbb{E}_{\mathbb{Q}}| = \sup_{f \in \mathcal{F}} |\mathbb{E}_{\hat{\mathbb{P}}} - \mathbb{E}_{\hat{\mathbb{Q}}} + \mathbb{E}_{\hat{\mathbb{Q}}} - \mathbb{E}_{\mathbb{Q}}|$$

$$\le \sup_{f \in \mathcal{F}} \left[ \left| \mathbb{E}_{\hat{\mathbb{P}}} - \mathbb{E}_{\hat{\mathbb{Q}}} \right| + \left| \mathbb{E}_{\hat{\mathbb{Q}}} - \mathbb{E}_{\mathbb{Q}} \right| \right] \quad (46)$$

$$\le \sup_{f \in \mathcal{F}} \left| \mathbb{E}_{\hat{\mathbb{P}}} - \mathbb{E}_{\hat{\mathbb{Q}}} \right| + \sup_{f \in \mathcal{F}} \left| \mathbb{E}_{\hat{\mathbb{Q}}} - \mathbb{E}_{\mathbb{Q}} \right|,$$

where $\hat{\mathbb{P}}$ denotes the empirical distribution of source tasks in the meta-training phase, $\hat{\mathbb{Q}}$ denotes the empirical distribution of the support set for target tasks in the meta-testing phase, and $\mathbb{Q}$ denotes the expected distribution of the query set.

For the *first term* in Eq.46, according to Lemma C.5 and Lemma C.6, we can bound the result as follows:

$$\sup_{f \in \mathcal{F}} \left| \mathbb{E}_{\hat{\mathbb{P}}} - \mathbb{E}_{\hat{\mathbb{Q}}} \right| \le 2\mathcal{R}(\mathcal{F}) + \sqrt{\frac{\log(1/\epsilon)}{2m}}$$

$$\le 2\sqrt{\frac{\nu \cdot \mathrm{rank}(\Sigma_{\mathrm{X}})}{m}} + \sqrt{\frac{\log(1/\epsilon)}{2m}}. \quad (47)$$

Similarly, for the *last term* in Eq.46, we can obtain the bounded result as follows:

$$\sup_{f \in \mathcal{F}} \left| \mathbb{E}_{\hat{\mathbb{Q}}} - \mathbb{E}_{\mathbb{Q}} \right| \le 2\mathcal{R}(\mathcal{F}) + \sqrt{\frac{\log(1/\epsilon)}{2n_q}}$$

$$\le 2\sqrt{\frac{\nu \cdot \mathrm{rank}(\Sigma_{\mathrm{X}})}{n_q}} + \sqrt{\frac{\log(1/\epsilon)}{2n_q}}. \quad (48)$$

Combining Eqs.47 and 48, we can obtain the desired results as shown in Theorem 5.3. Consequently,

$$\sup_{f \in \mathcal{F}} |\mathbb{E}_{\hat{\mathbb{P}}} - \mathbb{E}_{\mathbb{Q}}| \le \sup_{f \in \mathcal{F}} \left| \mathbb{E}_{\hat{\mathbb{P}}} - \mathbb{E}_{\hat{\mathbb{Q}}} \right| + \sup_{f \in \mathcal{F}} \left| \mathbb{E}_{\hat{\mathbb{Q}}} - \mathbb{E}_{\mathbb{Q}} \right|$$

$$\le 2\sqrt{\frac{\nu \cdot \mathrm{rank}(\Sigma_{\mathrm{X}})}{m}} + \sqrt{\frac{\log(1/\epsilon)}{2m}} + 2\sqrt{\frac{\nu \cdot \mathrm{rank}(\Sigma_{\mathrm{X}})}{n_q}} + \sqrt{\frac{\log(1/\epsilon)}{2n_q}}$$

$$= \left( 2\sqrt{\nu \cdot \mathrm{rank}(\Sigma_{\mathrm{X}})} + \sqrt{\frac{\log(1/\epsilon)}{2}} \right) \left( \sqrt{\frac{1}{m}} + \sqrt{\frac{1}{n_q}} \right). \quad (49)$$

$$\square$$

Thus, we complete the proof.

## D Statistics and Descriptions of Datasets

In this section, we provide detailed statistics and descriptions of the used datasets, which have been widely used in previous studies [5, 25, 50]. The detailed descriptions are provided below.

• **Amazon-Clothing** [28]: It is a product network constructed from the "Clothing, Shoes, and Jewelry" category on Amazon. In this dataset, each product is treated as a node, and its description is used to construct node features. A link is created between products if they are co-viewed. The labels are defined as the low-level product class. For this dataset, we use the 40/17/20 class split for meta-training/meta-validation/meta-testing.

• **CoraFull** [3]: It is a prevalent citation network. Each node represents a paper, and an edge is created between two papers if one cites the other. The nodes are labeled based on the topics of the papers. This dataset extends the previously widely used small dataset Cora by extracting raw data from the entire network. For this dataset, we use a 25/20/25 node class split for meta-training/meta-validation/meta-testing.

• **Amazon-Electronics** [28]: It is another Amazon product network that contains products belonging to the "Electronics" category. Each node represents a product, with its features representing the product description. An edge is created between products if there is a co-purchasing relationship. The low-level product categories are used as class labels. For this dataset, we use a 90/37/40 node category split for meta-training/meta-validation/meta-testing.

• **DBLP** [44]: It is a citation network where each node represents a paper, and the edges represent citation relationships between different papers. The abstracts of the papers are used to construct node features. The class labels of the nodes are defined as the publication venues of the papers. For this dataset, we use an 80/27/30 node category split for meta-training/meta-validation/meta-testing.

## E Descriptions of Baselines

In this section, we present the detailed descriptions of the selected baselines below.

### E.1 Traditional Meta-learning Method

**Protonet** [40]: It learns a metric space by acquiring prototypes of different categories from the support set and computes the similarity between the query samples and each prototype to predict their categories.

**MAML** [6]: It enables the meta-trainer to obtain a well-initialized parameter by performing one or more gradient update steps on the model parameters, allowing for rapid adaptation to downstream novel tasks with limited labeled data.

### E.2 Meta-learning with Fewer Tasks Method

**MetaMix** [58]: It enhances meta-training tasks by linearly combining the features and labels of samples from the support and query sets to improve the generalization of the model.

**MLTI** [60]: It generates additional tasks by randomly sampling a pair of tasks and interpolating their corresponding features and labels, replacing the original tasks for training.

**Meta-Inter** [19]: It proposes a domain-agnostic task augmentation method that utilizes expressive neural set functions to densify the distribution of meta-training tasks through a bi-level optimization process.

### E.3 Graph Meta-learning Method

**Meta-GNN** [65]: It seamlessly integrates MAML and GNNs in a straightforward manner, leveraging the MAML framework to acquire useful prior knowledge from previous tasks during the process of learning node embeddings, enabling it to rapidly adapt to novel tasks.

**GPN** [5]: It adopts the concept of Protonet for the few-shot node classification task. It uses a GNN-based encoder and evaluator to learn node embeddings and assess the importance of these nodes, while assigning novel samples to their closest categories.

**G-Meta** [15]: It constructs an individual subgraph for each node, transmits node-specific information within these subgraphs, and employs meta-gradients to learn transferable knowledge based on the MAML framework.

**Meta-GPS** [25]: It cleverly introduces prototype-based parameter initialization, scaling, and shifting transformations to better learn transferable meta-knowledge within the MAML framework and adapts to novel tasks more quickly.

**X-FNC** [51]: It first performs label propagation to obtain rich pseudo-labeled nodes based on Poisson learning, and then filters out irrelevant information through classifying nodes and an information bottleneck-based method to gather meta-knowledge across different meta-tasks with extremely supervised information.

**COSMIC** [53]: It proposes a contrastive meta-learning framework, which first explicitly aligns node embeddings by contrasting two-step optimization within each episode, and then generates hard node classes through a similarity-sensitive mixing strategy.

**TLP** [43]: It introduces the concept of transductive linear probing, initially pretraining a graph encoder through graph contrastive learning, and then applying it to obtain node embeddings during the meta-testing phase for downstream tasks.

**TEG** [17]: It designs a task-equivariant graph few-shot learning framework, leveraging equivariant neural networks to learn adaptive task-specific strategies, aimed at capturing task inductive biases to quickly adapt to unseen tasks.

### E.4 Implementation Details of Baselines

For traditional meta-learning models, we follow the same settings as [5, 25], and conduct careful hyperparameter search and report their optimal performance. For meta-learning with fewer tasks models, we uniformly use SGC as the graph encoder. Moreover, we adopt the following additional experimental settings. Specifically, for MetaMix, we allow it to perform task augmentation by generating the same number of nodes as those in the original support and query sets for each meta-training task. For MLTI and Meta-Inter, we make them to generate the same number of additional tasks as in our experiments to ensure fairness. For graph meta-learning baselines, we use the hyperparameters recommended in the original papers. All the experiments are conducted by NVIDIA 3090Ti GPUs with the Python 3.7 and PyTorch 1.13 environment.

## F More Experimental Results

### F.1 Model Performance with Sufficient Tasks

We present the experimental results of our method and other baselines in Tables S5 under sufficient tasks with the 5 way 5 shot setting. According to Table S5, we observe that the proposed SMILE achieves competitive performance compared to other baselines, thus providing strong evidence for its effectiveness in addressing the graph few-shot learning problem.

Also, we provide the results of these models in more challenging cross-domain experimental settings in Table S6. In this experimental setup, we first meta-train the model on the source domain and then evaluate it on the target domain. According to the results, similar to the in-domain ones, we find that our proposed approach still significantly outperforms all baselines, further demonstrating its ability to effectively extract transferable knowledge and exhibit strong generalization performance.

### F.2 Comparison of Performance with With-in Task Mixup and Increased Shot Numbers

Further, we conduct an interesting experiment to explore how our model augmented via within-task mixup fares against baselines enhanced by increasing shot count with external data. While our augmented data originates from within task distributions, we aim to evaluate its effectiveness compared to baselines explicitly inflated with external data. Specifically, we ran experiments using the 5-way 5-shot setting with five tasks on all datasets. As within-task mixup effectively doubles the shot count, we opt for the 5-way 10-shot setting for the baselines. Note that we have not included meta-learning with fewer tasks methods (*i.e.*, MetaMix, MLTI, and Meta-Inter) here, as they have already explicitly performed data augmentation or task augmentation. According to Table S7, we find that baseline models exhibit slight performance improvements from 5-way 5-shot to 5-way 10-shot settings, yet they still fail to outperform our model. This further highlights the superiority of SMILE.

**Table S5: Results (%) of different models with sufficient meta-training tasks under the 5-way 5-shot *in-domain* setting.**

| Model | Amazon-Clothing | | CoraFull | | Amazon-Electronics | | DBLP | |
|---|---|---|---|---|---|---|---|---|
| | Acc | F1 | Acc | F1 | Acc | F1 | Acc | F1 |
| Protonet | 63.51±3.62 | 63.70±2.59 | 55.65±3.76 | 52.92±3.66 | 59.72±2.69 | 61.50±2.62 | 56.32±2.39 | 55.39±2.32 |
| MAML | 66.12±3.12 | 67.82±2.92 | 56.52±2.70 | 55.39±3.15 | 59.02±3.49 | 58.31±3.20 | 49.93±3.62 | 47.79±3.16 |
| MetaMix | 83.19±2.95 | 82.12±2.56 | 70.36±2.39 | 68.59±2.69 | 78.25±3.25 | 77.09±3.11 | 80.26±2.55 | 79.06±2.59 |
| MLTI | 83.39±2.46 | 82.56±2.30 | 70.99±2.15 | 69.39±2.56 | 79.36±2.75 | 78.12±2.56 | 81.22±2.52 | 80.16±2.36 |
| Meta-Inter | 85.39±2.72 | 84.26±2.19 | 73.19±2.59 | 72.65±2.35 | 80.16±2.95 | 79.49±2.76 | 81.59±2.76 | 80.96±2.52 |
| Meta-GNN | 74.79±2.39 | 77.50±2.52 | 59.12±2.36 | 57.12±2.56 | 67.91±3.19 | 66.83±3.32 | 74.20±2.95 | 73.10±3.19 |
| GPN | 76.13±2.20 | 79.03±2.39 | 60.31±2.19 | 59.46±2.36 | 70.93±2.72 | 70.64±2.79 | 76.19±2.52 | 75.82±2.35 |
| G-Meta | 76.62±3.25 | 78.60±3.19 | 62.43±3.11 | 61.61±2.76 | 73.62±2.52 | 72.60±3.19 | 77.61±3.26 | 76.93±3.03 |
| Meta-GPS | 82.62±2.39 | 81.62±2.26 | 69.25±2.52 | 68.60±2.25 | 80.26±2.16 | 79.32±2.05 | 81.76±1.95 | 81.15±1.86 |
| X-FNC | 82.83±2.66 | 81.59±2.32 | 71.26±2.19 | 69.02±2.59 | 77.39±2.56 | 76.50±2.39 | 79.59±2.26 | 78.06±2.19 |
| COSMIC | 86.22±1.70 | 85.65±1.93 | 77.24±1.52 | 75.10±1.82 | 79.38±2.25 | 77.59±2.36 | 81.94±2.20 | 80.39±2.79 |
| TLP | 85.22±3.35 | 83.65±3.19 | 71.36±4.49 | 70.70±3.72 | 79.38±3.92 | 77.59±3.55 | 81.94±2.82 | 80.39±2.56 |
| TEG | **90.18±0.95** | **89.25±1.36** | **76.37±1.92** | **75.76±1.25** | **87.17±1.15** | **85.29±2.02** | 83.33±1.22 | 82.39±1.29 |
| SMILE | 88.86±1.12 | 88.59±1.16 | 75.50±1.26 | 75.14±1.39 | 85.55±1.62 | 84.95±1.29 | **83.90±1.19** | **83.42±1.56** |

**Table S6: Results (%) of different models with sufficient meta-training tasks under the 5-way 5-shot *cross-domain* setting.**

| Dataset | Amazon-Clothing→CoraFull | | CoraFull→Amazon-Clothing | | Amazon-Electronics→DBLP | | DBLP→Amazon-Electronics | |
|---|---|---|---|---|---|---|---|---|
| | Acc | F1 | Acc | F1 | Acc | F1 | Acc | F1 |
| Protonet | 36.46±3.19 | 22.85±2.72 | 36.52±2.93 | 33.36±2.90 | 42.76±2.78 | 39.30±2.72 | 37.86±2.66 | 29.47±2.62 |
| MAML | 34.01±3.39 | 20.95±3.12 | 35.19±2.96 | 34.29±2.60 | 38.47±3.10 | 32.51±3.19 | 35.92±2.70 | 26.70±3.06 |
| MetaMix | 40.16±2.13 | 38.25±2.39 | 44.19±2.56 | 42.39±2.92 | 53.10±2.42 | 51.90±2.35 | 47.36±2.56 | 45.19±2.32 |
| MLTI | 43.35±2.11 | 42.10±2.09 | 48.20±2.33 | 46.26±2.17 | 55.11±2.77 | 53.49±2.60 | 49.10±2.09 | 47.06±2.46 |
| Meta-Inter | 45.36±2.39 | 43.25±2.11 | 49.15±2.66 | 47.36±2.72 | 56.39±2.40 | 55.16±2.30 | 49.39±2.59 | 47.49±2.56 |
| Meta-GNN | 37.29±2.56 | 31.66±2.49 | 45.79±2.32 | 43.72±2.29 | 50.16±2.30 | 49.76±2.36 | 45.66±2.29 | 43.66±2.25 |
| GPN | 45.26±3.35 | 43.25±3.12 | 56.16±2.99 | 55.68±2.75 | 65.28±2.42 | 65.37±2.53 | 49.20±3.39 | 47.62±3.40 |
| G-Meta | 39.39±3.41 | 38.72±2.95 | 49.90±2.75 | 48.56±2.96 | 55.26±2.47 | 53.75±2.49 | 46.72±2.32 | 45.67±2.29 |
| Meta-GPS | 41.29±2.16 | 40.79±2.12 | 58.62±2.25 | 57.29±2.20 | 60.12±2.06 | 59.73±2.02 | 49.39±2.15 | 47.96±2.12 |
| X-FNC | 42.56±2.75 | 41.19±2.46 | 55.39±2.49 | 54.29±2.37 | 61.55±2.32 | 60.92±2.74 | 49.21±2.51 | 46.55±2.39 |
| COSMIC | 46.55±2.45 | 44.29±2.42 | 57.26±2.39 | 56.22±2.40 | 63.59±2.98 | 62.29±2.93 | 51.22±2.86 | 50.35±2.78 |
| TLP | 44.76±3.47 | 43.29±3.32 | 57.95±2.91 | 56.36±2.77 | 64.52±2.73 | 63.22±2.49 | 49.51±2.36 | 46.55±2.72 |
| TEG | 40.19±1.26 | 39.96±1.66 | 50.23±2.53 | 48.29±2.06 | 46.35±2.65 | 45.26±2.72 | 41.25±1.93 | 40.59±1.60 |
| SMILE | **49.08±1.23** | **47.46±1.42** | **62.72±2.02** | **61.29±1.86** | **68.96±1.12** | **68.03±1.06** | **53.38±1.32** | **52.70±1.25** |

## F.3 Model Performance with Alternative Across-task Mixup

To further demonstrate the superiority of our proposed across-task mixup, we attempt to replace it with the mixup strategy in the MLTI [60]. It directly performs mixup on the support set and query set from two tasks to achieve task augmentation, formally defined as:

$$\tilde{X}^s_{t;\tilde{k}} = \lambda X^s_{i;k} + (1-\lambda) X^s_{j;k'}, \quad \tilde{X}^q_{t;\tilde{k}} = \lambda X^q_{i;k} + (1-\lambda) X^q_{j;k'}. \quad (50)$$

We present the results with different mixup strategies under the 5-way 5-shot in-domain and cross-domain settings varying number of tasks in Table S8. Here, "*alternate*" denotes the model variant that performs the aforementioned task augmentation.

According to the results, we can conclude that the adopted across-task augmentation strategy consistently outperforms the task augmentation of MLTI on various experimental settings. One plausible reason is that the adopted prototype-based across-task mixup can generate more reliable data compared to instance-based one of other models, thus further reducing adverse oscillations when predicting examples beyond the training set.

**Table S7: Results of our model on the 5-way 5-shot 5 tasks in-domain setting compared with those of other models on the 5-way 10-shot 5 tasks setting.**

| Model | Amazon-Clothing | | CoraFull | | Amazon-Electronics | | DBLP | |
|---|---|---|---|---|---|---|---|---|
| | Acc | F1 | Acc | F1 | Acc | F1 | Acc | F1 |
| Protonet | 49.27±3.19 | 48.72±2.79 | 38.95±2.61 | 37.87±2.68 | 50.14±2.73 | 49.80±2.65 | 50.91±2.56 | 49.99±2.73 |
| MAML | 46.42±2.30 | 45.16±2.34 | 40.12±2.35 | 38.79±2.56 | 39.95±2.62 | 38.60±2.71 | 45.26±2.42 | 43.92±2.93 |
| Meta-GNN | 57.06±3.72 | 53.19±3.47 | 43.90±3.12 | 42.19±3.09 | 45.16±2.19 | 43.96±2.15 | 52.35±2.35 | 51.76±2.32 |
| GPN | 68.64±2.73 | 68.39±2.07 | 45.22±3.45 | 43.29±3.21 | 54.23±2.73 | 53.17±2.50 | 70.39±2.29 | 70.13±2.65 |
| G-Meta | 61.29±2.59 | 60.96±2.32 | 46.72±2.40 | 45.35±2.49 | 45.20±3.12 | 43.56±3.22 | 53.99±2.93 | 49.20±2.71 |
| Meta-GPS | 63.22±2.35 | 60.36±2.29 | 52.16±2.19 | 50.10±2.11 | 49.32±2.95 | 46.02±2.62 | 58.19±1.98 | 56.22±1.72 |
| X-FNC | 70.22±2.97 | 69.35±2.73 | 56.99±2.65 | 53.96±2.42 | 62.19±2.34 | 59.39±2.47 | 71.66±2.39 | 70.92±2.91 |
| COSMIC | 77.29±2.93 | 76.33±2.82 | 64.22±2.53 | 62.33±2.45 | 65.25±2.46 | 64.18±2.72 | 72.10±3.16 | 71.15±3.06 |
| TLP | 73.21±2.29 | 72.13±2.12 | 53.11±2.35 | 52.01±2.19 | 64.51±2.77 | 63.18±3.02 | 72.39±3.03 | 71.35±3.12 |
| TEG | 80.22±2.12 | 79.36±2.36 | 64.16±1.76 | 63.31±1.63 | 66.06±1.96 | 65.16±1.95 | 74.36±2.03 | 73.19±2.39 |
| SMILE | **82.80±1.32** | **82.49±1.52** | **66.34±1.29** | **65.70±1.56** | **67.30±1.20** | **66.30±1.19** | **75.88±1.29** | **75.05±1.36** |

**Table S8: Results (%) of different model variants on the datasets under various experimental settings.**

| Task | Model | Amazon-Clothing | | CoraFull | | Amazon-Electronics | | DBLP | |
|---|---|---|---|---|---|---|---|---|---|
| | | Acc | F1 | Acc | F1 | Acc | F1 | Acc | F1 |
| 5 | *alternate* | 81.69±1.39 | 81.52±1.22 | 65.42±1.70 | 64.79±1.35 | 65.96±1.19 | 65.20±1.22 | 74.92±1.32 | 73.95±1.15 |
| | ours | **82.80±1.32** | **82.49±1.52** | **66.34±1.29** | **65.70±1.56** | **67.30±1.20** | **66.30±1.19** | **75.88±1.29** | **75.05±1.36** |
| 10 | *alternate* | 82.22±1.50 | 82.02±1.12 | 70.79±1.26 | 70.22±1.32 | 69.99±1.02 | 69.16±1.39 | 75.26±1.70 | 73.90±1.96 |
| | ours | **83.46±1.66** | **82.88±1.35** | **71.72±1.95** | **71.15±1.76** | **70.76±1.06** | **70.05±1.09** | **76.64±1.22** | **75.77±1.19** |
| 15 | *alternate* | 82.99±1.29 | 82.39±1.36 | 69.72±1.52 | 69.03±1.32 | 72.25±1.39 | 71.12±1.47 | 78.52±1.56 | 77.35±1.30 |
| | ours | **83.92±1.16** | **83.33±1.22** | **70.78±1.59** | **70.19±1.42** | **73.48±1.36** | **72.66±1.22** | **79.56±1.26** | **78.77±1.76** |
| 20 | *alternate* | 83.72±1.38 | 82.95±1.49 | 71.92±1.71 | 71.72±1.96 | 74.39±1.86 | 74.32±1.72 | 79.52±1.39 | 78.51±1.77 |
| | ours | **84.66±1.55** | **84.52±1.39** | **72.60±1.66** | **72.10±1.55** | **75.42±1.52** | **75.42±1.29** | **80.50±1.72** | **79.61±1.55** |
| Task | Model | Amazon-Clothing→CoraFull | | CoraFull→Amazon-Clothing | | Amazon-Electronics→DBLP | | DBLP→Amazon-Electronics | |
| | | Acc | F1 | Acc | F1 | Acc | F1 | Acc | F1 |
| 5 | *alternate* | 42.02±2.36 | 41.42±1.72 | 55.37±1.97 | 54.36±1.86 | 61.30±1.79 | 61.02±1.75 | 45.14±1.39 | 43.72±1.55 |
| | ours | **42.64±2.02** | **41.27±1.65** | **56.36±2.02** | **55.25±1.75** | **62.44±1.35** | **61.66±1.29** | **46.24±1.60** | **44.54±1.62** |
| 10 | *alternate* | 43.59±1.22 | 41.79±1.59 | 57.99±1.73 | 56.76±1.62 | 63.12±1.79 | 62.29±2.25 | 47.95±2.22 | 46.82±2.16 |
| | ours | **45.14±1.29** | **43.69±1.10** | **58.84±1.79** | **57.53±1.56** | **64.54±1.22** | **64.16±1.20** | **48.82±1.12** | **47.26±1.19** |
| 15 | *alternate* | 45.12±1.77 | 43.39±1.56 | 58.16±1.32 | 54.76±1.39 | 63.70±1.42 | 63.26±1.49 | 48.39±1.51 | 46.89±1.60 |
| | ours | **45.88±1.36** | **44.10±1.22** | **59.08±1.55** | **55.96±1.50** | **65.04±1.19** | **64.43±1.26** | **49.26±1.55** | **47.70±1.36** |
| 20 | *alternate* | 46.12±1.09 | 44.26±1.75 | 57.99±1.79 | 57.56±1.73 | 63.92±1.60 | 63.22±1.92 | 47.28±1.50 | 47.06±1.35 |
| | ours | **46.72±1.96** | **45.65±1.66** | **59.38±1.62** | **58.25±1.60** | **65.78±1.32** | **65.42±1.26** | **49.52±1.29** | **47.88±1.26** |

## F.4 Impact of Original Tasks

Our proposed model is trained during the meta-training stage utilizing a merged task $\mathcal{D}_{all}$ composed of original tasks $\mathcal{D}_{org}$ and interpolated tasks $\mathcal{D}_{aug}$. In contrast, MLTI disregards the use of original tasks and solely uses the generated new tasks for training. However, this approach raises concerns as the model does not directly encounter the data distribution of the original tasks, potentially compromising its generalization ability. Moreover, in scenarios where training data is already scarce, this practice results in the wastage of valuable data resources. To further quantitatively explore the impact of these original tasks on model performance, we conduct additional experiments with several model variants under different experimental settings across all the datasets. Here, "*w/o original taks*" denotes that we exclude the original tasks and solely rely on the generated tasks for model training. All the results are presented in Table S9.

Based on the above results, the training strategy utilized consistently brings about performance improvements compared to solely

**Table S9: Results (%) of different training methods on the datasets.**

| Task | Model | Amazon-Clothing | | CoraFull | | Amazon-Electronics | | DBLP | |
|---|---|---|---|---|---|---|---|---|---|
| | | Acc | F1 | Acc | F1 | Acc | F1 | Acc | F1 |
| 5 | *w/o original tasks* | 81.74±1.79 | 81.60±1.69 | 65.69±1.75 | 64.72±1.52 | 65.36±1.22 | 64.20±1.29 | 74.05±1.36 | 73.09±1.19 |
| | ours | **82.80±1.32** | **82.49±1.52** | **66.34±1.29** | **65.70±1.56** | **67.30±1.20** | **66.30±1.19** | **75.88±1.29** | **75.05±1.36** |
| 10 | *w/o original tasks* | 82.06±1.55 | 81.36±1.19 | 70.29±1.32 | 69.95±1.36 | 69.09±1.42 | 68.16±1.33 | 75.36±1.75 | 72.90±1.73 |
| | ours | **83.46±1.66** | **82.88±1.35** | **71.72±1.95** | **71.15±1.76** | **70.76±1.06** | **70.05±1.09** | **76.64±1.22** | **75.77±1.19** |
| 15 | *w/o original tasks* | 82.93±1.32 | 81.99±1.56 | 69.92±1.31 | 68.26±1.37 | 72.29±1.19 | 71.20±1.40 | 78.39±1.52 | 77.20±1.35 |
| | ours | **83.92±1.16** | **83.33±1.22** | **70.78±1.59** | **70.19±1.42** | **73.48±1.36** | **72.66±1.22** | **79.56±1.26** | **78.77±1.76** |
| 20 | *w/o original tasks* | 83.69±1.39 | 83.26±1.42 | 71.90±1.52 | 70.72±1.94 | 74.29±1.26 | 73.29±1.71 | 79.49±1.93 | 78.39±1.78 |
| | ours | **84.66±1.55** | **84.52±1.39** | **72.60±1.66** | **72.10±1.55** | **75.42±1.52** | **75.42±1.29** | **80.50±1.72** | **79.61±1.55** |

| Task | Model | Amazon-Clothing→CoraFull | | CoraFull→Amazon-Clothing | | Amazon-Electronics→DBLP | | DBLP→Amazon-Electronics | |
|---|---|---|---|---|---|---|---|---|---|
| | | Acc | F1 | Acc | F1 | Acc | F1 | Acc | F1 |
| 5 | *w/o original tasks* | 41.32±2.09 | 40.56±1.71 | 55.29±1.67 | 53.76±1.66 | 61.26±1.73 | 60.10±1.69 | 45.15±1.32 | 43.69±1.59 |
| | ours | **42.64±2.02** | **41.27±1.65** | **56.36±2.02** | **55.25±1.75** | **62.44±1.35** | **61.66±1.29** | **46.24±1.60** | **44.54±1.62** |
| 10 | *w/o original tasks* | 43.56±1.29 | 41.66±1.53 | 57.29±1.66 | 56.65±1.60 | 63.09±1.39 | 62.16±2.12 | 47.32±2.23 | 46.92±2.11 |
| | ours | **45.14±1.29** | **43.69±1.10** | **58.84±1.79** | **57.53±1.56** | **64.54±1.22** | **64.16±1.20** | **48.82±1.12** | **47.26±1.19** |
| 15 | *w/o original tasks* | 44.23±1.27 | 43.92±1.67 | 57.90±1.34 | 55.29±1.31 | 64.29±1.44 | 63.20±1.40 | 48.29±1.39 | 46.29±1.63 |
| | ours | **45.88±1.36** | **44.10±1.22** | **59.08±1.55** | **55.96±1.50** | **65.04±1.19** | **64.43±1.26** | **49.26±1.55** | **47.70±1.36** |
| 20 | *w/o original tasks* | 45.66±1.11 | 43.69±1.72 | 57.97±1.73 | 57.22±1.60 | 64.26±1.35 | 63.22±1.46 | 48.46±1.59 | 46.29±1.23 |
| | ours | **46.72±1.96** | **45.65±1.66** | **59.38±1.62** | **58.25±1.60** | **65.78±1.32** | **65.42±1.26** | **49.52±1.29** | **47.88±1.26** |

training the model on generated new tasks. This further underscore the effectiveness of our employed training method.

## F.5 Impact of Graph Augmentation

In this work, we actually employ feature-level mixup. We would like to explain our rationale. Firstly, mixup is a simple and highly effective technique that aligns with the straightforward concept we aim to convey in our research. Secondly, by employing feature-level mixup, we can directly address the scarcity of nodes and tasks within a given task by enriching both the node and task distributions. Given that the target data is graph-structured data, someone may wonder how the model performance would be affected by utilizing other augmentation methods designed for graph-structured data instead of mixup. We argue that using graph augmentation methods would primarily impact the learning of node representations and would have minimal influence on subsequent algorithms that extract generalizable knowledge from limited data. To verify our hypothesis, we utilize GAug [64] and GMix [54] for graph augmentation on the vanilla SGC [55] node representation learning module. Subsequently, we directly conduct metric-based few-shot node classification without introduced dual-level mixup techniques. The corresponding model variants are denoted as "w GAug" and "w GMix". Moreover, "SGC" denotes that we directly utilize SGC for node representation learning and subsequently perform metric-based few-shot node classification. The experimental results are presented in Table S10.

Based on the results, we observe that incorporating graph augmentation techniques can result in slight performance improvements for the SGC model. This is attributed to the ability of graph augmentation to facilitate the learning of high-quality node representations by the model.

## F.6 Model Performance with Different Graph Encoders

To further validate the flexibility of our proposed model, we replace the graph backbone with GCN [18], GAT [47], SAGE [12], and GraphGPS [37] under the 5 way 5 shot setting. The results of these experiments are presented in Table S11. According to the results, we find that our proposed method, even when equipped with different graph encoders, still achieves excellent performance across various datasets under different experimental settings, providing strong evidence of its effectiveness.

## F.7 Model Performance for Few-shot Graph Classification

In this section, we explore the application of our method to few-shot graph classification tasks. By utilizing graph pooling operations to obtain graph-level features, extending our model to downstream graph-level tasks is straightforward. To support this, we select several representative datasets Letter-high, ENZYMES, TRIANGLES, and Reddit, which are widely used for few-shot graph classification. We provide the statistics of evaluated datasets in Table S12. Detailed descriptions of these datasets are provided below.

• **Letter-high** contains graphs representing the English alphabet, with each label corresponding to a specific letter type.

• **ENZYMES** is a protein tertiary structure dataset composed of enzymes from the BRENDA database, with each class corresponding to a top-level enzyme.

**Table S10: Results (%) of different models on the datasets.**

| Task | Model | Amazon-Clothing | | CoraFull | | Amazon-Electronics | | DBLP | |
|---|---|---|---|---|---|---|---|---|---|
| | | Acc | F1 | Acc | F1 | Acc | F1 | Acc | F1 |
| 5 | SGC | 69.02±1.94 | 67.12±1.65 | 43.02±1.75 | 42.29±1.50 | 50.20±1.26 | 47.92±1.23 | 70.11±1.33 | 69.25±1.15 |
| | w GAug | 69.22±1.29 | 67.55±1.35 | 43.56±2.29 | 42.39±2.17 | 50.26±1.74 | 47.99±1.70 | 70.39±1.59 | 69.39±1.24 |
| | w GMix | 70.26±1.51 | 68.36±1.26 | 45.29±1.17 | 44.02±1.39 | 51.92±1.56 | 49.39±1.39 | 71.30±1.32 | 70.70±1.20 |
| | ours | **82.80±1.32** | **82.49±1.52** | **66.34±1.29** | **65.70±1.56** | **67.30±1.20** | **66.30±1.19** | **75.88±1.29** | **75.05±1.36** |
| 10 | SGC | 71.29±2.25 | 70.32±2.19 | 46.11±1.73 | 44.72±1.95 | 52.10±2.13 | 53.56±2.36 | 74.39±1.94 | 73.52±1.75 |
| | w GAug | 71.66±2.02 | 70.62±2.13 | 46.29±1.97 | 44.99±1.92 | 52.19±2.12 | 53.66±2.49 | 74.56±2.03 | 73.92±1.59 |
| | w GMix | 72.90±1.90 | 71.36±1.75 | 48.09±2.05 | 45.92±2.13 | 53.59±2.06 | 54.02±2.31 | 75.30±2.19 | 74.12±2.37 |
| | ours | **83.46±1.66** | **82.88±1.35** | **71.72±1.95** | **71.15±1.76** | **70.76±1.06** | **70.05±1.09** | **76.64±1.22** | **75.77±1.19** |
| 15 | SGC | 72.12±1.62 | 71.49±1.51 | 51.96±1.36 | 50.20±1.35 | 55.02±1.39 | 54.12±1.47 | 76.22±1.57 | 75.12±1.65 |
| | w GAug | 72.53±1.79 | 71.99±2.05 | 52.06±1.49 | 50.36±1.70 | 55.29±1.96 | 54.20±1.71 | 76.59±2.13 | 75.26±2.03 |
| | w GMix | 73.56±1.72 | 72.26±1.59 | 54.29±1.77 | 51.20±1.67 | 56.49±1.72 | 55.39±1.40 | 77.29±1.93 | 76.12±2.14 |
| | ours | **83.92±1.16** | **83.33±1.22** | **70.78±1.59** | **70.19±1.42** | **73.48±1.36** | **72.66±1.22** | **79.56±1.26** | **78.77±1.76** |
| 20 | SGC | 72.66±1.79 | 72.16±1.49 | 56.35±1.57 | 55.32±1.92 | 57.52±1.56 | 56.69±1.75 | 76.99±1.91 | 76.52±1.72 |
| | w GAug | 73.06±2.02 | 72.36±1.99 | 56.25±1.97 | 55.29±1.72 | 57.36±1.96 | 56.49±1.47 | 76.90±2.20 | 76.02±2.14 |
| | w GMix | 73.66±1.95 | 72.99±1.73 | 57.29±1.60 | 56.92±1.95 | 59.90±2.09 | 58.20±2.03 | 77.32±1.76 | 77.20±1.65 |
| | ours | **84.66±1.55** | **84.52±1.39** | **72.60±1.66** | **72.10±1.55** | **75.42±1.52** | **75.42±1.29** | **80.50±1.72** | **79.61±1.55** |

| Task | Model | Amazon-Clothing→CoraFull | | CoraFull→Amazon-Clothing | | Amazon-Electronics→DBLP | | DBLP→Amazon-Electronics | |
|---|---|---|---|---|---|---|---|---|---|
| | | Acc | F1 | Acc | F1 | Acc | F1 | Acc | F1 |
| 5 | SGC | 36.22±2.03 | 35.15±1.75 | 41.06±1.62 | 40.13±1.60 | 59.99±1.72 | 58.91±1.63 | 43.02±1.71 | 43.02±1.54 |
| | w GAug | 36.42±2.17 | 35.56±1.92 | 41.26±1.49 | 40.36±1.73 | 60.36±1.49 | 59.10±1.70 | 43.19±1.56 | 42.69±1.44 |
| | w GMix | 37.52±2.11 | 36.26±2.04 | 40.39±2.19 | 40.02±2.11 | 60.99±1.75 | 60.36±1.59 | 43.32±1.74 | 42.79±1.96 |
| | ours | **42.64±2.02** | **41.27±1.65** | **56.36±2.02** | **55.25±1.75** | **62.44±1.35** | **61.66±1.29** | **46.24±1.60** | **44.54±1.62** |
| 10 | SGC | 39.39±1.52 | 38.52±1.11 | 42.01±1.37 | 41.45±1.32 | 61.92±1.41 | 61.42±1.72 | 47.19±2.21 | 46.36±2.01 |
| | w GAug | 39.76±1.79 | 38.62±1.56 | 42.29±1.66 | 41.66±1.73 | 62.09±1.57 | 61.69±1.63 | 47.30±2.19 | 46.62±1.94 |
| | w GMix | 39.92±1.93 | 39.36±1.76 | 43.39±2.03 | 42.91±1.97 | 62.39±1.60 | 62.02±1.49 | 47.39±1.75 | 46.92±1.66 |
| | ours | **45.14±1.29** | **43.69±1.10** | **58.84±1.79** | **57.53±1.56** | **64.54±1.22** | **64.16±1.20** | **48.82±1.12** | **47.26±1.19** |
| 15 | SGC | 41.29±2.56 | 40.76±2.35 | 43.76±2.02 | 42.66±2.43 | 63.12±1.97 | 62.39±1.74 | 47.12±3.19 | 45.26±3.05 |
| | w GAug | 41.53±1.22 | 40.99±1.60 | 43.96±1.31 | 42.96±1.22 | 63.29±1.40 | 62.90±1.45 | 47.39±1.32 | 45.69±1.63 |
| | w GMix | 42.36±1.41 | 41.26±2.73 | 44.39±2.59 | 43.20±2.44 | 63.39±2.40 | 63.09±2.37 | 47.59±2.18 | 45.92±2.10 |
| | ours | **45.88±1.36** | **44.10±1.22** | **59.08±1.55** | **55.96±1.50** | **65.04±1.19** | **64.43±1.26** | **49.26±1.55** | **47.70±1.36** |
| 20 | SGC | 41.92±3.26 | 41.16±2.72 | 45.02±3.73 | 44.66±3.60 | 63.90±3.35 | 63.19±2.66 | 47.36±2.59 | 45.62±2.23 |
| | w GAug | 42.06±3.19 | 41.60±2.74 | 45.26±2.59 | 44.99±2.73 | 64.06±2.56 | 63.42±2.70 | 47.96±2.19 | 45.99±2.31 |
| | w GMix | 42.36±2.56 | 41.99±2.40 | 45.99±2.37 | 45.26±2.42 | 63.92±2.31 | 63.29±2.56 | 48.36±2.15 | 47.92±2.26 |
| | ours | **46.72±1.96** | **45.65±1.66** | **59.38±1.62** | **58.25±1.60** | **65.78±1.32** | **65.42±1.26** | **49.52±1.29** | **47.88±1.26** |

• **TRIANGLES** consists of graphs, where the category is determined by the number of triangles (3-cliques) present in each graph.
• **Reddit** contains graphs representing threads, where each node represents a user, and different graph labels correspond to different types of forums.

We choose several representative few-shot graph classification models, GSM [4], AS-MAML [27], FAITH [52], and SMART [23], for comparison. The detailed descriptions of these models are presented below.
• **GSM** [4]: It generates a set of superclasses through graph spectral metrics and constructs corresponding super-graphs to model the relationships between the classes.

• **AS-MAML** [27]: It directly combines GNN and MAML to quickly adapt to unseen test graphs, utilizing a step controller to enhance the robustness of the meta-trainer.
• **FAITH** [52]: It captures task relevance by constructing hierarchical graphs of varying granularity, thereby enhancing the model's adaptability to unseen new classes.
• **SMART** [23]: It replaces the complex meta-learning training paradigm with a simpler transfer learning approach, utilizing graph contrastive learning and prompt learning to enhance the model's representation extraction capability and learning efficiency.

We present the results of our model and these baselines in the Table S13. According to the results, we can find that our model significantly surpasses other baseline models across multiple datasets

**Table S11: Results (%) of our model on the datasets under various backbones.**

| Task | Backbone | Amazon-Clothing | | CoraFull | | Amazon-Electronics | | DBLP | |
|------|----------|-----|-----|-----|-----|-----|-----|-----|-----|
| | | Acc | F1 | Acc | F1 | Acc | F1 | Acc | F1 |
| 5 | GCN | 81.76±1.39 | 81.29±1.95 | 65.96±1.66 | 65.25±2.02 | **67.96±2.11** | **66.82±1.96** | 75.52±2.32 | 74.75±2.30 |
| | GAT | 80.19±2.21 | 79.59±2.02 | 63.99±1.97 | 63.36±1.73 | 66.75±1.66 | 65.32±1.55 | 73.96±1.50 | 73.96±1.45 |
| | SAGE | **82.92±1.70** | **82.52±1.49** | **66.82±1.53** | **66.36±1.39** | 67.10±1.75 | 66.16±1.50 | 75.39±2.22 | 74.95±1.92 |
| | GraphGPS | 80.25±1.99 | 78.26±1.97 | 64.39±1.85 | 63.95±1.82 | 66.19±1.80 | 65.29±2.36 | 73.29±2.39 | 72.72±2.52 |
| | SGC (ours) | 82.80±2.19 | 82.49±2.26 | 66.34±2.28 | 65.70±2.52 | 67.30±2.59 | 66.30±2.32 | **75.88±2.21** | **75.05±2.22** |
| 10 | GCN | 82.56±2.36 | 82.29±2.30 | **72.20±2.15** | **71.35±2.45** | 69.76±2.29 | 69.22±2.26 | 75.39±2.17 | 74.62±2.79 |
| | GAT | 81.92±2.36 | 80.96±2.66 | 70.92±1.80 | 70.39±1.77 | 68.15±2.16 | 67.29±2.30 | 74.99±2.22 | 74.29±2.15 |
| | SAGE | 83.02±1.73 | 82.56±1.79 | 71.96±2.32 | 71.52±2.25 | 70.26±2.26 | 70.15±2.21 | 76.32±2.46 | 75.32±2.39 |
| | GraphGPS | 81.26±2.11 | 80.76±2.23 | 70.15±2.25 | 69.69±2.76 | 67.92±2.82 | 66.95±2.37 | 74.25±2.29 | 74.12±2.46 |
| | SGC (ours) | **83.46±1.66** | **82.88±1.35** | 71.72±1.95 | 71.15±1.76 | **70.76±1.32** | **70.05±1.90** | **76.64±1.56** | **75.77±1.38** |

| Task | Model | Amazon-Clothing→CoraFull | | CoraFull→Amazon-Clothing | | Amazon-Electronics→DBLP | | DBLP→Amazon-Electronics | |
|------|-------|-----|-----|-----|-----|-----|-----|-----|-----|
| | | Acc | F1 | Acc | F1 | Acc | F1 | Acc | F1 |
| 5 | GCN | 42.16±1.88 | 41.75±1.82 | 55.79±1.81 | 54.66±1.85 | 62.19±1.39 | 60.99±1.22 | 46.09±1.26 | 45.69±1.21 |
| | GAT | 40.25±1.29 | 40.02±1.26 | 53.32±1.22 | 52.12±1.34 | 61.02±1.36 | 60.19±1.42 | 45.62±1.58 | 45.19±1.52 |
| | SAGE | 41.99±1.57 | 40.95±1.59 | 55.96±1.64 | 55.29±1.43 | **62.92±1.41** | **62.29±1.30** | 47.19±1.36 | 45.20±1.33 |
| | GraphGPS | 39.62±1.49 | 38.66±1.52 | 52.29±1.55 | 51.96±1.66 | 60.72±1.60 | 59.79±1.53 | 45.02±1.42 | 44.16±1.51 |
| | SGC (ours) | **42.64±1.20** | **41.27±1.22** | **56.36±1.39** | **55.25±1.30** | 62.44±1.55 | 61.66±1.26 | 46.24±1.22 | 44.54±1.66 |
| 10 | GCN | 43.12±1.24 | 42.26±1.26 | 57.65±1.32 | 57.16±1.39 | 63.25±1.42 | 62.35±1.49 | 47.22±1.55 | 46.29±2.12 |
| | GAT | 43.19±2.12 | 42.59±2.19 | 56.99±2.22 | 56.36±2.55 | 62.39±2.36 | 61.96±1.91 | 46.29±2.16 | 45.56±2.59 |
| | SAGE | 44.66±2.66 | 44.25±2.75 | 58.16±2.30 | 57.25±2.36 | 63.92±2.15 | 62.92±2.50 | 48.09±1.72 | 47.02±1.76 |
| | GraphGPS | 42.26±1.79 | 42.09±1.92 | 55.26±1.22 | 55.12±2.05 | 61.52±2.17 | 60.92±2.09 | 46.25±2.01 | 45.29±2.06 |
| | SGC (ours) | **45.14±2.19** | **43.69±2.22** | **58.84±2.39** | **57.53±2.26** | **64.54±1.79** | **64.16±1.86** | **48.82±1.79** | **47.22±1.66** |

**Table S12: Statistics of the evaluated datasets.**

| Dataset | # Graphs | # Nodes | # Edges | # Classes | # Novel |
|---------|----------|---------|---------|-----------|---------|
| Letter-high | 2,250 | 4.67 | 4.50 | 15 | 4 |
| ENZYMES | 600 | 32.63 | 62.14 | 6 | 2 |
| TRIANGLES | 2,000 | 20.85 | 35.50 | 10 | 3 |
| Reddit | 1,111 | 391.41 | 456.89 | 11 | 4 |

under various experimental settings, clearly demonstrating the superiority and adaptability of our approach.

### F.8 Performance Varies with Mixup Parameters

In this section, we investigate the sensitivity of the performance of our proposed model to the parameters of the Beta distribution used in the developed mixing strategy. We present how the performance of our model varies with the Beta distribution parameters, $\alpha$ and $\beta$, across different datasets under the 5-way 5-shot 5 tasks few-shot experimental settings. For simplicity, we always keep $\alpha$ and $\beta$ equal. As shown in Fig. S1, we observe that our model exhibits good robustness concerning this parameter. As the parameters change, the model performance maintains a stable trend.

### F.9 Exploring Model Generalization Gap

Moreover, we further investigate whether SMILE can improve its generalization capability by reducing the generalization gap, where the generalization gap is empirically defined as the disparity between the model's accuracy on the meta-training tasks and its accuracy on the meta-testing tasks. Fig. S2 illustrates the generalization gap induced by different models under the 5-way 5-shot few-shot setting, including both in-domain and cross-domain settings. Upon comparing the disparities depicted in Fig. S2 (a) and (b), it is evident that the discrepancy between the training and testing accuracies when employing our method consistently remains smaller than that of other methods. These results empirically support our theoretical findings, showing that, compared to standard training without dual-levl mixup, SMILE consistently exhibits a smaller generalization gap with high probability. This further confirms the effectiveness of our proposed method under both in-domain and cross-domain settings.

## G Visualization Study

To visually present the introduced dual-level mixup strategy, we leverage t-SNE [46] to visualize the results of dual-level mixup on the Amazon-clothing dataset under the 5-way 5-shot with 5 tasks few-shot setting, as shown in Fig. S3. Specifically, in the within-task mixup, we randomly select one task consisting of support and query sets. In the across-task, we interpolate 50 tasks, where the task embeddings are the average of the contained node embeddings. According to Fig. S3, we observe that the interpolated nodes within each task and the interpolated tasks generated by SMILE indeed densify the node and task distributions, thereby enhancing the model generalization capability.

**Table S13: Results of different models on few-shot graph classification tasks.**

| Model | Letter-high | | ENZYMES | | TRIANGLES | | Reddit | |
|---|---|---|---|---|---|---|---|---|
| | 5-shot | 10-shot | 5-shot | 10-shot | 5-shot | 10-shot | 5-shot | 10-shot |
| GSM | 69.91±5.90 | 73.28±3.64 | 55.42±5.74 | 60.64±3.84 | 71.40±4.34 | 75.60±3.67 | 41.59±4.12 | 45.67±3.68 |
| AS-MAML | 69.44±0.75 | 75.93±0.53 | 49.83±1.12 | 52.30±1.43 | 78.42±0.67 | 80.39±0.56 | 36.96±0.74 | 41.47±0.83 |
| FAITH | 71.55±3.58 | 76.65±3.26 | 57.89±4.65 | 62.16±4.11 | 79.59±4.05 | 80.79±3.53 | 42.71±4.18 | 46.63±4.01 |
| SMART | 74.17±2.75 | 76.89±1.55 | 59.80±3.39 | 65.11±2.70 | 79.39±2.45 | 80.43±2.12 | 43.83±2.21 | 47.75±2.77 |
| Ours | **76.56±1.96** | **79.22±2.32** | **61.35±3.75** | **67.19±2.89** | **81.22±2.66** | **82.56±2.75** | **45.72±2.39** | **49.26±2.82** |

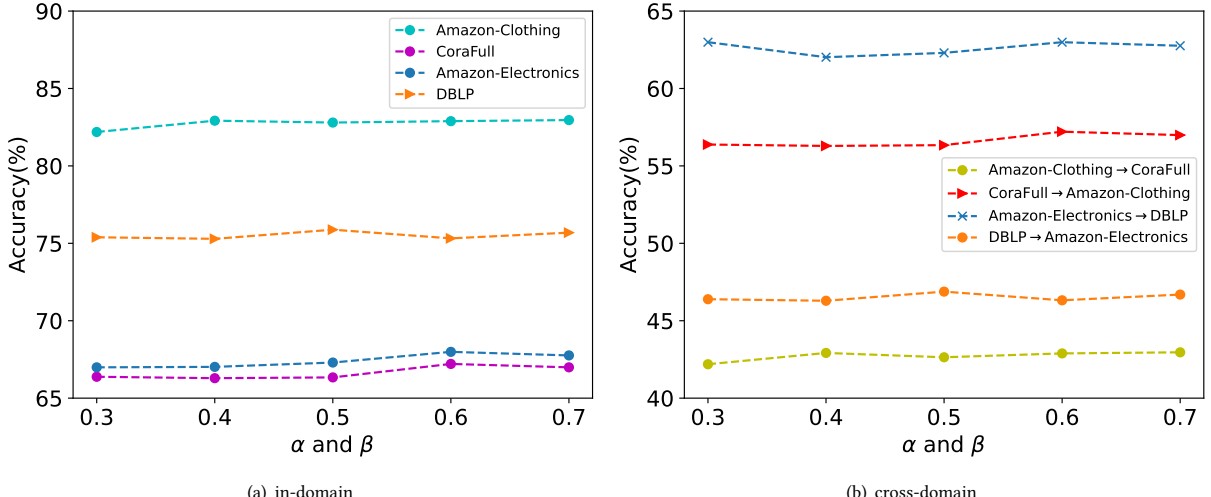

(a) in-domain

(b) cross-domain

**Figure S1: Model performance varies with the values of the mixup hyperparameters $\alpha$ and $\beta$.**

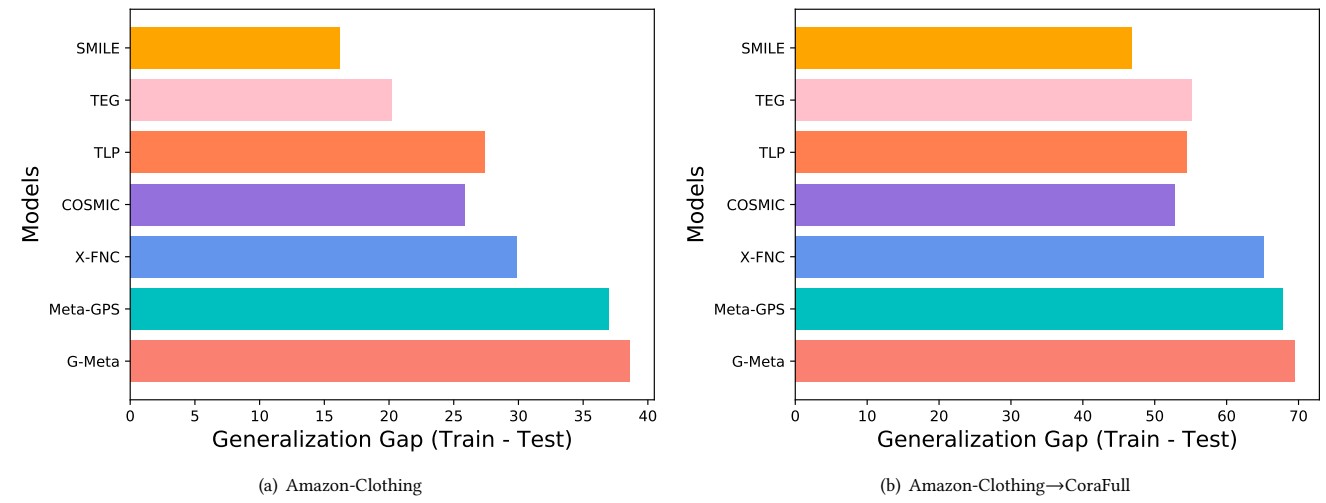

(a) Amazon-Clothing

(b) Amazon-Clothing→CoraFull

**Figure S2: (a) Generalization gap over several methods on the Amazon-Clothing dataset under the 5-way 5-shot 5 tasks setting. (b) Generalization gap over several methods on the Amazon-Clothing dataset under the 5-way 5-shot 5 tasks setting.**

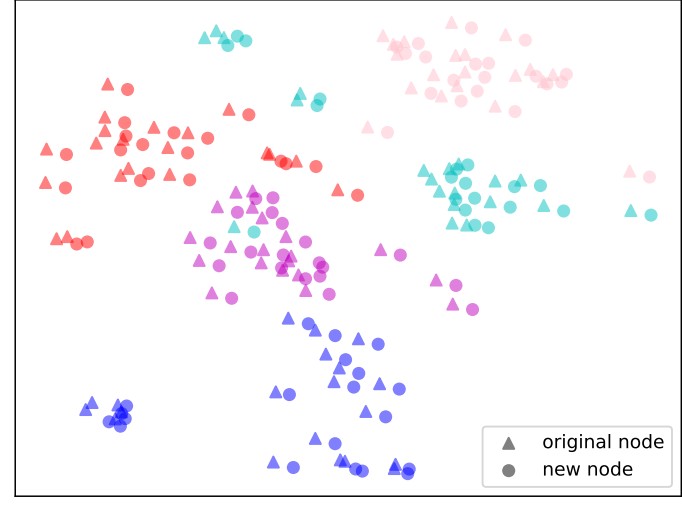

(a) within-task mixup

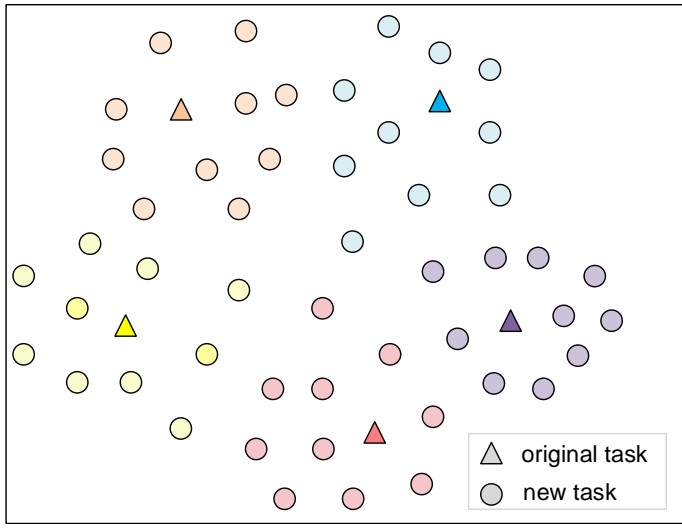

(b) across-task mixup

**Figure S3: Visualization of the dual-level mixup strategies. In (a), the original nodes in each task are represented by triangles, while the generated nodes are represented by circles, with colors indicating the corresponding classes. In (b), the original tasks are represented by triangles, the generated tasks are represented by circles, and the colors indicate the most similar original tasks.**

