# OpenReview forum: "Dual-level Mixup for Graph Few-shot Learning with Fewer Tasks"
_ACM.org/TheWebConf/2025/Conference — WWW 2025 Poster_

### Official Review · Reviewer_SQza · 2024-11-07

**Novelty:** 4
**Technical Quality:** 3

**Review:**

The paper introduces SMILE designed to address the challenge of graph few-shot learning in scenarios where the number of tasks is limited. SMILE employs a dual-level mixup strategy that enriches both the node and task distributions, enhancing the model's ability to generalize from limited data.

**Questions:**

Pros:

1. The dual-level mixup strategy is a creative solution to the few-task problem in graph few-shot learning.

2. The paper provides extensive experimental results demonstrating the effectiveness of SMILE across multiple datasets.

 Cons:

1. The introduction of a dual-level mixup may increase the complexity of the model, which could be a drawback in scenarios where simplicity is preferred.

2. The performance of SMILE may be sensitive to the parameters of the Beta distribution used in the mixup process, requiring careful tuning.

3. The paper does not discuss the explainability of the model's decisions.

4. While the paper uses mixup for data augmentation, the impact of other graph augmentation techniques on the model's performance is not explored.


Questions:

1. What is the computational cost associated with the dual-level mixup strategy compared to traditional few-shot learning methods?

2. How robust is SMILE to noisy data, including noisy node features and edges?

3. How does SMILE handle class imbalance, which is common in few-shot learning scenarios?

**Reviewer Confidence:**

2: The reviewer is willing to defend the evaluation, but it is likely that the reviewer did not understand parts of the paper

**Scope:**

3: The work is somewhat relevant to the Web and to the track, and is of narrow interest to a sub-community

---

### Official Review · Reviewer_Gr8k · 2024-11-21

**Novelty:** 5
**Technical Quality:** 5

**Review:**

Strength
1. The authors propose the SMILE for graph few-shot learning with fewer tasks, which includes node representation Learning and dual-level mixup.
2. The authors analyze the proposed method theoratically, and prove its generalization.

Weakness
1. The meaning of some mathematical symbols are ambiguous. For example, what does $\tilde{X}^{s}_{t,Q,\tilde{k}}$ represent in Eq. 5?
2. The meaning of some shapes in Figure 2 is unclear. For example, what do the triangles, stars, and circles represent?

**Questions:**

1. Whether the models used for node representation and score evaluation in the node representation learning module require training?
2. Whether all of meta-tasks include continuous labels? Can discrete labels be interpolated using the dual-level mixup strategy to generate more samples and tasks?

**Reviewer Confidence:**

3: The reviewer is confident but not certain that the evaluation is correct

**Scope:**

3: The work is somewhat relevant to the Web and to the track, and is of narrow interest to a sub-community

---

### Official Review · Reviewer_b3kg · 2024-11-26

**Novelty:** 4
**Technical Quality:** 5

**Review:**

This study focuses on few-show learning with fewer tasks, and proposes an approaches leveraging a dual-level mixup strategy. It also gives a solid theoretically analysis to reveal the generalization capabilities and complexity of the approach. The extensive experiments show the effectiveness of the approach.

S1: The experiments are well-conducted with comprehensive evaluation of the number of tasks, the ablation study, the choice of mixup, the impact of original tasks, and the model parameters. The proposed approach performs well on the accuracy and the F1 score.
S2: The design of the approach has solid theoretical analyses and full explanations. Theoretical claims are supported with experiments.
S3: The presentation is clear and the work is reproducible. The approach and the experimental explanation are well illustrated.

W1: The references [42, 51, 59]  are all recent works in the few-shot learning on graphs. The reason why the approach does not compare with them in the experimental study needs to be given.
W2: The result on Tables 2 and 3 are based on the pairwise t-test with 95% confidence. It seems that the proposed approach does not show a large margin (as illustrated in the Abstract) with other competitive models across all datasets. It is also shown in different ways from Tables S5-S11.
W3: The datasets are all product networks and citation networks. It would be better to validate the effectiveness of the approach in other fields.

[42] Zhen Tan, Ruocheng Guo, Kaize Ding, and Huan Liu. 2023. Virtual node tuning for few-shot node classification. In SIGKDD.
[50] Song Wang, Kaize Ding, Chuxu Zhang, Chen Chen, and Jundong Li. 2022. Taskadaptive few-shot node classification. In SIGKDD. 1910–1919.
[59] Huaxiu Yao, Chuxu Zhang, YingWei, Meng Jiang, SuhangWang, Junzhou Huang, Nitesh Chawla, and Zhenhui Li. 2020. Graph few-shot learning via knowledge transfer. In AAAI.

**Questions:**

Q1: Why the approach does not compare with the references [42, 51, 59] (please refer to W1)?


Q2: How to show a large margin (as illustrated in the Abstract) between the proposed approach and the other competitive models (please refer to W2)?

Q3: How about the efficiency of the proposed approach?

**Reviewer Confidence:**

3: The reviewer is confident but not certain that the evaluation is correct

**Scope:**

4: The work is relevant to the Web and to the track, and is of broad interest to the community

---

### Official Review · Reviewer_HxTv · 2024-11-29

**Novelty:** 4
**Technical Quality:** 5

**Review:**

This paper introduces SMILE that effectively addresses challenges arising from the scarcity of labeled tasks and limited support samples in few-shot scenarios by employing a dual-level mixup strategy. The author demonstrates theoretical support through a reduction in generalization error and empirical superiority across multiple datasets in both in-domain and cross-domain settings.

Strengths
1. The dual-level mixup strategy is innovative in its application to graph few-shot learning, addressing task and sample scarcity issues.
2. The explicit utilization of node degree information for representation learning introduces a novel way to enhance graph embeddings.
3. Strong empirical performance across datasets, with theoretical guarantees.

Weaknesses
1. The proposed method relies on node degree information as prior knowledge during node representation learning. However, the authors do not discuss why node degree is particularly effective or whether other graph structural features (e.g., clustering coefficient, node centrality) were considered. Could this focus on node degree limit the method's generalizability to other graph types?
2. The cross-domain experiments lack a detailed analysis of the domain shift between the source and target domains. For instance, what specific statistical or structural differences exist between domains, and how do these differences challenge the model? Without such context, it is difficult to fully appreciate SMILE's claimed robustness to distribution shifts.

**Questions:**

1. In addition to the reported cross-domain experiments, did the authors quantify the degree of distribution shift (e.g., using KL divergence or Wasserstein distance) between source and target domains? If not, would such an analysis help clarify SMILE’s robustness under varying levels of domain shift?

2. In the cross-domain experiments, could the authors provide more details about the differences between the source and target domains? For example, how do node features and graph structures differ? How might these differences impact SMILE’s performance compared to baselines?

3. As far as I understand, the earliest Meta-GNN methods utilized both SGC and GCN frameworks for node representation learning. Could the authors clarify which framework was specifically used in the proposed method?

**Reviewer Confidence:**

4: The reviewer is certain that the evaluation is correct and very familiar with the relevant literature

**Scope:**

3: The work is somewhat relevant to the Web and to the track, and is of narrow interest to a sub-community

---

### Official Review · Reviewer_5DJX · 2024-11-29

**Novelty:** 5
**Technical Quality:** 5

**Review:**

This paper aims to address the problem of meta-learning based graph few-shot learning by proposing a novel method called SMILE (SiMple yet effectIve approach for graph few-shot Learning with fEwer tasks). The proposed approach introduces a dual-level mixup strategy to mitigate the challenge of limited labeled data and tasks, enhancing the generalization ability of graph neural networks.

# Pros
1. Novelty: The dual-level mixup strategy, encompassing both within-task and across-task mixup, is an innovative approach to address the issue of few-shot learning in graphs. This is particularly relevant in scenarios where obtaining a large number of meta-training tasks is infeasible.

1. Effectiveness: The empirical results demonstrate that SMILE consistently outperforms existing competitive models across various datasets, including both in-domain and cross-domain settings. This indicates the robustness and applicability of the proposed method.

1. Theoretical Justification: The authors provide a theoretical analysis showing how SMILE enhances the model's generalization ability by regularizing model weights and constricting the upper bound of generalization error.

1. Utilization of Node Degrees: Incorporating node degrees as prior information to enhance node representations is a practical and effective strategy, contributing to the model's overall performance.

# Cons
1. Complexity of Implementation: While the proposed method is described as simple, the dual-level mixup strategy may add a layer of complexity to implementation. Detailed guidance or pseudocode would be beneficial for reproducibility.

1. Evaluation Metrics: Although the paper shows significant performance improvements, it would be useful to provide more detailed evaluation metrics, such as precision, recall, and F1-score, to give a comprehensive view of the model's performance.

1. Task Generation: The process of generating new tasks through across-task mixup might introduce synthetic noise, which could affect the quality of the training tasks. A discussion on how to mitigate potential negative effects would be helpful.

1.  Comparison with Baselines: While the paper compares SMILE with several baseline models, including GPN, G-Meta, and Meta-GPS, it would be useful to include a broader range of baselines to strengthen the comparative analysis.

**Questions:**

1. How do you ensure the quality of synthetic tasks generated through the across-task mixup, and what measures are in place to mitigate any potential noise introduced?
1. Have you considered comparing SMILE with a broader range of baseline models to strengthen the comparative analysis?

**Ethics Review Flag:**

Yes

**Reviewer Confidence:**

3: The reviewer is confident but not certain that the evaluation is correct

**Scope:**

4: The work is relevant to the Web and to the track, and is of broad interest to the community